



# Characterising Methane Gas and Environmental Response of the Figaro Taguchi Gas Sensor (TGS) 2611-E00

Adil Shah[1], Olivier Laurent[1], Luc Lienhardt[1], Grégoire Broquet[1], Rodrigo Rivera Martinez[1], Elisa Allegrini[2], Philippe Ciais[1]

[1]Laboratoire des Sciences du Climat et de l'Environnement (CEA-CNRS-UVSQ), Institut Pierre-Simon Laplace, Université Paris-Saclay, Site de l'Orme des Merisiers, 91191 Gif-sur-Yvette, France
[2]SUEZ Smart Solutions, 15-27 Rue de Port, 92000 Nanterre, France

*Correspondence to*: Adil Shah (adil.shah@lsce.ipsl.fr)

**Abstract.** In efforts to improve methane source characterisation, networks of cheap high frequency in situ sensors are required,
with a parts-per-million level methane mole fraction ([$CH_4$]) precision. Low-cost semiconductor-based metal oxide sensors, such as the Figaro Taguchi Gas Sensor (TGS) 2611-E00, may satisfy this requirement. The resistance of these sensors decreases in response to the exposure of reducing gases, such as methane. In this study, we set out to characterise the Figaro TGS 2611-E00, in efforts to eventually yield [$CH_4$] when deployed in the field. We found that different gas sources, containing the same ambient 2 ppm [$CH_4$] level, yielded different resistance responses. For example, synthetically generated air
containing 2 ppm [$CH_4$] produced a lower sensor resistance than 2 ppm [$CH_4$] found in natural ambient air, due to possible interference from supplementary reducing gas species in ambient air, though the specific cause of this phenomenon is not clear. TGS 2611-E00 carbon monoxide response is small and incapable of causing this effect. For this reason, ambient laboratory air was selected as a testing gas standard, to naturally incorporate such background effects into a reference resistance. Figaro TGS 2611-E00 resistance is sensitive to temperature and water mole fraction ([$H_2O$]). Therefore, a reference
resistance using this ambient air gas standard was characterised for five sensors (each inside its own field logging enclosure) using a large environmental chamber, where logger enclosure temperature ranged between 8° C and 38° C and [$H_2O$] ranged between 0.4% and 1.9%. [$H_2O$] dominated resistance variability in the standard gas. A linear [$H_2O$] and temperature model fit was derived, resulting in a root-mean-squared error (RMSE) between measured and modelled resistance in standard gas of between ±0.4 kΩ and ±1.0 kΩ for the five sensors, corresponding to a fractional resistance uncertainty of less than ±3% at
25° C and 1% [$H_2O$]. The TGS 2611-E00 loggers were deployed at a landfill site for 242 days before and 96 days after sensor testing. Yet the standard (*i.e* ambient air) reference resistance model fit based on temperature and [$H_2O$] could not replicate resistance measurements made in the field, where [$CH_4$] was mostly expected to be close to the ambient background, with minor enhancements. This field disparity may have been due to variability in sensor cooling dynamics, a difference in ambient air composition during environmental chamber testing compared to the field or variability in natural sensor response, either
spontaneously or environmentally driven. Despite difficulties in replicating a standard reference resistance in the field, we devised an excellent methane characterisation model up to 1 000 ppm [$CH_4$], using the ratio between measured resistance with [$CH_4$] enhancement and a reference resistance in standard gas. A bespoke power-type fit between resistance ratio and [$CH_4$]



resulted in a RMSE between modelled and measured resistance ratio of no more than $\pm 1\% \ \Omega \ \Omega^{-1}$ for the five sensors. This fit and it corresponding fit parameters were then inverted and the original resistance ratio values were used to derive [CH$_4$],

yielding an inverted model [CH$_4$] RMSE of less than $\pm 1$ ppm, where [CH$_4$] was limited to 28 ppm. Our methane response model allows other reducing gases to be included if necessary, by characterising additional model coefficients. Our model shows that a 1 ppm [CH$_4$] enhancement above the ambient background results in a resistance drop of between 1.4% and 2.0%, for the five tested sensors. With future improvements in deriving a standard reference resistance, the TGS 2611-E00 offers great potential in measuring [CH$_4$] with a parts-per-million precision.

## 1. Introduction

Methane (CH$_4$) is a potent greenhouse gas (Mitchell, 1989) with many poorly characterised sources (Jackson *et al*, 2020). Yet as atmospheric methane mole fraction ([CH$_4$]) is increasing (Rigby et al., 2007, Nisbet et al., 2014), improved source flux quantification is required (Saunois et al., 2016, Nisbet et al., 2019, Turner et al., 2019). This necessitates improvements in fast-response (less than 1 minute) and high frequency (at least 0.1 Hz) in situ [CH$_4$] sampling. CH$_4$ is a trace gas with a low natural

ambient atmospheric background (defined to be $(2\pm1)$ ppm hereon), which is two orders of magnitude lower than carbon dioxide mole fraction ([CO$_2$]) (Dlugokencky et al., 1994, Dlugokencky, 2022).

Fast-response in situ [CH$_4$] sampling techniques span many capabilities and costs (Hodgkinson and Tatam, 2013, Schuyler and Guzman, 2017). The best measurements are achieved using tuneable infrared (IR) lasers (Baer et al., 2002, Frish, 2014), but cheaper broad-band IR can also be used in techniques such as non-dispersive IR spectroscopy (Hummelgård *et al*, 2015),

at expense of precision (Shah et al., 2019). Alternatively, semiconductor-based metal oxide (SMO) sensors have been available for several decades (Fleischer and Meixner, 1995, Barsan et al., 2007, Reinelt et al., 2017, Ponzoni et al., 2017). Though they are marketed for low-precision applications, their sub-$10^2$€ cost (Eugster and Kling, 2012, Riddick et al., 2020) merits a thorough assessment of their fast-response [CH$_4$] sampling capability (Collier-Oxandale et al., 2018, Honeycutt et al., 2019).

SMO sensor resistance is influenced by gas exposure (Kohl 1990). For n-type sensors containing metal lattices in their most

oxidised state (Kohl, 2001), oxygen surface chemisorption forms $O^{2-}$, $O_2^-$ or $O^-$ (depending on temperature), thus decreasing near-surface electron density in the conduction band (Barsan et al., 2007, Das et al., 2014). This catalyses SMO surface oxidation of reducing gases, thereby releasing electrons into the conduction band to lower resistance (Kohl, 1989, Ponzoni et al., 2017). For CH$_4$, this initially produces a hydrogen atom and methyl radical (Kohl, 1989), before eventual formation of carbon dioxide (CO$_2$) and water (Suto and Inoue, 2010, Chakraborty et al., 2006, Glöckler et al., 2020).

n-type SMO sensors may contain tin, vanadium or zinc oxides (Hong et al., 2020). As tin oxides (SnO$_x$) are poorly CH$_4$-selective (Kim et al., 1997, Collier-Oxandale et al., 2018), catalysts may be introduced (Hong et al., 2020). Noble metals such as platinum (Pt) and palladium (Pd) influence sensitivity and selectivity (Kohl, 1990, Xue et al., 2019), often by catalysing



oxygen dissociation (Kim et al., 1997, Navazani et al., 2020, Wang et al., 2010). For example, Haridas and Gupta (2013) improved $CH_4$ detection by uniformly applying Pd clusters to $SnO_x$, whereas Suto and Inoue (2010) employed a Pt-black

catalyst layer, to block hydrogen and carbon monoxide (CO). This yielded ±0.004 ppm [$CH_4$] agreement with a high-precision reference (HPR) instrument in background conditions (Suto and Inoue, 2010). Elsewhere, Yang et al. (2020) printed zeolite film on their Pd-loaded $SnO_x$ sensor, to catalytically oxidise CO and ethanol.

Most SMO sensors contain packed grains (Ponzoni et al., 2017, Hong et al., 2020), with sufficient touching grains to facilitate bulk conduction (Kohl, 2001). Smaller grains or more pores amplify surface area and thus, sensitivity (Wang et al., 2010).

This was achieved by Kim et al. (1997) who mixed $SnO_x$ powder with alumina or silica supported noble metals (detecting 500 ppm [$CH_4$]). Some SMO sensors instead utilise films (Suto and Inoue, 2010, Haridas and Gupta, 2013, Yang et al., 2020), for example Moalaghi et al., (2020) applied $SnO_x$ layers on alumina chips, whereas Chakraborty et al. (2006) painted iron-doped $SnO_x$ layers on alumina tubes. The Chakraborty et al. (2006) sensor exhibited peak 1 000 ppm $CH_4$ sensitivity at 350° C, but peak 1 000 ppm butane sensitivity at 425° C (depending on Pd content). Xue et al. (2019) printed a Pt flower pattern on

silicon dioxide film, for maximal surface area. Zhang et al. (2019) decorated 2% $SnO_x$ on uniform hexagonal nickel oxide sheets in their p-type $CH_4$ sensor, to optimise sensitivity and selectivity. Gagaoudakis et al., (2020) developed a transparent 100 nm thick polycrystalline p-type nickel oxide sensor, using aluminium. However, ultraviolet radiation was required to restore resistance, after gas exposure (Gagaoudakis et al., 2020).

Nanotubes and graphene structures may alternatively be used (Ponzoni et al., 2017, Hong et al., 2020) for better surface

adsorption (Navazani et al., 2020). Kooti et al. (2019) tested one-dimensional nanoscale rods, to be mixed with porous graphene nanosheets, where $CH_4$ could diffuse into the small pores, improving selectivity. Navazani et al. (2020) made an $SnO_x$ sensor 28 times more $CH_4$-sensitive (at 100 ppm), by combining it with Pt-doped multi-walled carbon nanotubes. Elsewhere, Das et al. (2014) used 2.4 nm $SnO_x$ quantum dots to detect as little as 50 ppm [$CH_4$]. A high surface to volume ratio and quantum effects enabled low-temperature (150° C) $CH_4$ sensitivity (Das et al., 2014).

Most SMO sensors operate at up to 400° C (Barsan et al., 2007), to enable oxygen vacancies to diffuse into the bulk material (Kohl, 1990). Airflow may consequently cause indirect sensor effects (Eugster et al., 2020). Cooler 150° C sensors have been developed, for example by Das et al. (2014) or by Kooti et al. (2019), which detected down to 1 000 ppm [$CH_4$]. Elsewhere, Xue et al. (2019) sampled 500 ppm [$CH_4$] with their 100° C sensor. Room temperature sensors have also been trailed (Navazani et al., 2020), for example, Haridas and Gupta (2013) developed a sensor using ultraviolet radiation to generate photo-induced

oxygen ions. This improved 200 ppm $CH_4$ sensitivity by three orders of magnitude (Haridas and Gupta, 2013). Conversely, Moalaghi et al. (2020) developed a hot (700° C up to 850° C) $SnO_x$ thermal decomposition sensor, to theoretically detect 50 ppm [$CH_4$]. $CH_4$ thermal stability enhanced its selectivity compared to hydrogen and CO (Moalaghi et al., 2020).



Water also influences SMO sensors (Collier-Oxandale et al., 2019, Navazani et al., 2020, Rivera Martinez et al., 2021) by competing for oxygen absorption sites (Kohl, 1989) at the expense of sensitivity (Wang et al., 2010, Yang et al., 2020). This effect may be temperature-dependent, whereby heat enhances water desorption (Kohl, 2001). While dry sampling may resolve this (Kohl, 1989, Suto and Inoue, 2010, Sasakawa et al., 2010), some sensors require wet air for normal operation (Eugster and Kling, 2012, Riddick et al., 2020).

Following robust physical sensor characterisation, empirical gas testing may then be performed in preparation for field deployment (Kim et al., 1997, Barsan et al., 2007, Honeycutt et al., 2019, Zhang et al., 2019, Daugela et al., 2020). A field-ready SMO sensor includes a sensitive layer, a substrate, electrodes (Barsan et al., 2007, Kooti et al. 2019, Glöckler et al., 2020) and a logger (Ferri et al., 2009, Collier-Oxandale et al., 2018). Concurrent logging of environmental conditions is invaluable (van den Bossche et al., 2017, Daugela et al., 2020, Cho et al., 2022). As an example of actual field application, Sasakawa et al. (2010) deployed nine Suto and Inoue (2010) sensors in Siberian wetlands. Thanks to regular calibrations, [CH4] measurements contributed towards regional surface flux emission estimates (Sasakawa et al., 2010). Gonzalez-Valencia et al. (2014) mapped landfill surface fluxes using flux chambers containing a suite of IR and SMO sensors. Daugela et al. (2020) used Hanwei Electronics Co., Ltd. MQ2 and MQ4 sensors, to crudely localise landfill emission hotspots. Honeycutt et al. (2021) utilised MQ4 sensors within a sampling network for autonomous deployment, with a 1 000 ppm [CH4] targeted detection limit. Kim et al. (2021) exploited low SMO sensor mass for unmanned aerial vehicle deployment, to derive landfill CH4 hotspots and surface fluxes. The sensor was laboratory-tested up to a maximum [CH4] of 200 ppm (Kim et al., 2021).

Figaro Engineering Inc. (Mino, Osaka, Japan) produce fast-response grain-based SMO sensors (Ferri et al., 2009, Eugster and Kling, 2012), which are more stable than the MQ4 (Honeycutt et al., 2019). Figaro sensors require wet air for normal operation (Rivera Martinez et al., 2021), thereby ruling out dry calibrations (Riddick et al., 2020). Eugster and Kling (2012) therefore performed Figaro Taguchi Gas Sensor (TGS) 2600 field characterisation with an HPR, over an Arctic lake. CO cross-sensitivity caused complications (Eugster and Kling, 2012), as encountered by Collier-Oxandale et al. (2018), elsewhere. The TGS 2600 sensor is also hydrogen-sensitive (Ferri et al., 2009). Eugster et al. (2020) yielded ±0.1 ppm model agreement with an HPR, from 7 years of background [CH4] Arctic sampling. Riddick et al. (2020) deployed the TGS 2600 for 3 months at a gas extraction site, sampling up to a 6 ppm [CH4] maximum, with ±0.01 ppm measurement uncertainty, following laboratory HPR characterisation.

Collier-Oxandale et al. (2019) combined a Figaro TGS 2600 and TGS 2602 (non-CH4) sensor to improve CH4 selectively and to combat cross-sensitivities. A subset of field sampling was used for HPR training, with the remainder for model testing (Collier-Oxandale et al., 2019). Casey et al. (2019) applied a similar field HPR training and testing approach to ten packages containing various sensors (including a TGS 2600 and TGS 2602), which were deployed across an oil and gas extraction region. Linear and artificial neural network (ANN) models were both able to derive [CH4], but correlated gas emissions from the same source may have confounded model output in this multi-sensor approach (Casey et al., 2019). Eugster et al. (2020)



also tested an ANN model, which performed better in warmer conditions. Rivera Martinez et al. (2021) used 47 days of TGS 2600, TGS 2611-C00 and TGS 2611-E00 sampling to derive background [$CH_4$] with ANN models. 70% of sampling was used for HPR training, typically resulting in less than ±0.2 ppm root-mean-squared error (RMSE), but the position of the 30% testing window effected model performance (Rivera Martinez et al., 2021). Elsewhere, Rivera Martinez et al. (2022) produced laboratory-generated methane spikes of between 3 ppm and 24 ppm over 130 days, which were sampled by four different TGS

2611-C00 and TGS 2611-E00 loggers. 70% of the data was used to train linear, polynomial and ANN models to replicate the spikes, using an HPR, with a target RMSE of ±2 ppm (Rivera Martinez et al., 2022).

The Figaro TGS 2611-E00 is more $CH_4$-selective as it incorporates a CO filter (van den Bossche et al., 2017, Bastviken et al., 2020, Figaro Engineering Inc., 2021), at the expense of $CH_4$ sensitivity (Eugster et al., 2020). van den Bossche et al. (2017) tested a TGS 2611-E00 in background [$CH_4$] for 31 days, following laboratory calibration, resulting in −1 ppm accuracy and

±1.7 ppm precision. Cho et al. (2022) sampled simulated gas leaks using 19 TGS 2611-E00 units, for four days, applying a universal laboratory calibration to all sensors, with a 100 ppm targeted detection limit. Jørgensen et al. (2020) sampled up to 90 ppm [$CH_4$] while HPR field testing a TGS 2611-E00 for 100 hours on the Greenland Ice Sheet, resulting in ±1.69 ppm RMSE. It then sampled autonomously for 18 days (Jørgensen et al., 2020). Bastviken et al. (2020) tested various TGS 2611-E00 calibration models up to 700 ppm [$CH_4$], for use in surface flux chambers. Sieczko et al. (2020) deployed TGS 2611-E00

flux chambers over three boreal lakes to characterise $CH_4$ emission variability. Although they calibrated each sensor, strong diurnal environmental outcomes were inferred from this imprecise sensor (Sieczko et al., 2020).

Due to its superior $CH_4$ selectivity, we characterised the TGS 2611-E00, with the eventual objective of measuring [$CH_4$] during outdoor field deployment. In order to derive [$CH_4$] with confidence, we conducted a series of robust laboratory characterisation tests, to understand the core principles of sensor response to various external factors. Our sensor characterisation approach was

thoroughly tested using 338 days of field sampling. Two logging systems were used, as described in Sect. 2: one for autonomous field sampling and the other for controlled testing of multiple sensors. Our overall characterisation process is outlined in Fig. 1. As a first step, sensor response to different standard gas samples was characterised, in the absence of $CH_4$ enhancements (see Sect. 3.1 and Sect. 3.2). Water mole fraction ([$H_2O$]) and temperature response were then characterised in a large environmental chamber in Sect. 3.3. A specific [$CH_4$] enhancement model fit was derived in Sect. 3.4. Sensor CO, $CO_2$

and oxygen response were also tested (see Sect. 3.5, Sect. 3.6 and Sect. 3.7). Then, to test sensor applicability in field conditions, ten sensors were deployed at a landfill site, providing a prolonged dataset with which to test our characterisation approach. [$H_2O$] and temperature measurements were used to model field resistance for five of these sensors, for comparison with actual resistance measurements (see Sect. 4). The quality of the environmental resistance model fit is discussed in Sect. 5 and we summarise our outcomes in Sect. 6.



## 2. Materials and logging methods

### 2.1 Sensor overview

Here we describe the basic operating principles of the Figaro TGS 2611-E00, referred to hereafter as "Figaro", unless otherwise stated. The Figaro is an SMO sensor, sensitive to hydrogen and light hydrocarbons (including $CH_4$), featuring an incorporated CO and ethanol filter (Figaro Engineering Inc., 2021). The Figaro internal heater and SMO element both operate at a $(5.0\pm0.2)$ V supply voltage ($V_s$). Figaro resistance ($R$) responds to surrounding gas exposure, which can be inferred by measuring the precise voltage drop ($V_d$) across a resistor of fixed load resistance ($R_l$), connected in series with the Figaro sensor electrodes (see Fig. 2), using Eq. (1) (Collier-Oxandale et al., 2018).

$$R = R_l \cdot \left( \frac{V_s}{V_d} - 1 \right) \tag{1}$$

$V_d$ is effectively used to gauge current flow, thereby quantifying resistance at a set $V_s$. $R_l$ may take a minimum value of 0.45 k$\Omega$ (Figaro Engineering Inc., 2021). However, for maximal sensitivity, $R_l$ should be selected to target a similar order of magnitude to $R$, depending on the sensor type and the predicted sampling conditions. A higher $R_l$ permits better sensitivity at lower $[CH_4]$, but limits precision when detecting larger enhancements.

### 2.2 Field logging system

To measure Figaro resistance in the field, we used ten Systematic Observations of Facility Intermittent Emissions (SOOFIE) logging systems (referred to hereafter as System A), manufactured by Scientific Aviation, Inc (Boulder, Colorado, USA). The ten systems are labelled from LSCE001 to LSCE010 (see Fig. 3, for example). Each system enclosure includes a Figaro sensor, connected in series with a 5 k$\Omega$ load resistor. Air is drawn towards the Figaro, using a downwards facing fan, in a similar style to Cho et al. (2022). An SHT85 environmental sensor (Sensirion AG, Staefa, Switzerland) records System A temperature ($T_A$) and relative humidity. The system is powered by a 12 V rechargeable lithium-ion phosphate battery, connected to a solar panel. This is converted to a stable Figaro 5 V power supply on an internal circuit board. An Arduino data logger records minute-average $V_d$, $T_A$ and relative humidity measurements, which are wirelessly transmitted to an Internet server using a cellular network board inside each box, similar to Honeycutt et al. (2021). Three systems (LSCE005, LSCE006 and LSCE007) also transmit minute-average wind speed and direction measurements from their own two-dimensional Gill WindSonic anemometers (Gill Instruments Ltd., Lymington, Hampshire, UK), connected to each of the three System A enclosures.

### 2.3 Laboratory testing logging system

A bespoke laboratory logger was designed, with five sockets, to facilitate simultaneous Figaro testing (referred to hereafter as System B). The 0.1 dm$^3$ cell has a glass exterior with a stainless steel head (see Fig. 4), which was adapted from a filter (FS-



2K-D, M&C TechGroup Germany GmbH, Ratingen, Germany). Each Figaro socket is connected in series with a high-precision $(5.00 \pm 0.05)$ k$\Omega$ load resistor (Vishay Intertechnology, Inc., Malvern, Pennsylvania, USA). 18 bit analogue-to-digital
converter chips (MCP3424, Microchip Technology Inc., Chandler, Arizona, USA) measure 1 Hz $V_d$ for each Figaro. This chip is ready-mounted onto an ADC Pi board (Apexweb Ltd, Swanage, Dorset, UK), which is connected to a Raspberry Pi 3B+ logging computer (Raspberry Pi Foundation, Cambridge, UK), using similar software to Rivera Martinez et al. (2021). A raw ADC Pi board $V_s$ measurement is recorded, alongside raw Figaro $V_d$, to linearly calibrate the ADC Pi. Furthermore, a ground reference offset correction between the Figaro sensors and the ADC Pi board is applied to $V_d$.

Preliminary tests with a single power supply yielded unstable $V_d$ measurements, as background activity on the logging computer influences total current draw. $V_s$ also influences Figaro CH$_4$ sensitivity (see Appendix A). Therefore, the logging computer and Figaro power supplies are split, with a common ground, as suggested elsewhere (van den Bossche et al., 2017, Daugela et al., 2020). A high-precision power supply unit (T3PS23203P, Teledyne LeCroy Inc., Chestnut Ridge, New York, USA) provides Figaro power, with a supply voltage accuracy of at least 35 mV.

An SHT85 sensor measures System B temperature and relative humidity at 1 Hz inside the cell. In addition, the Figaro cell outlet is fed through towards a Picarro G2401 gas analyser (Picarro, Inc., Santa Clara, California, USA), serving as an HPR. It records [CH$_4$], [H$_2$O], carbon monoxide mole fraction ([CO]) and [CO$_2$] at a maximum sampling frequency of 0.2 Hz, although this frequency declines depending on the complexity of the gas mixture. The Picarro G2401 offers sampling with a high temporal stability (Yver Kwok et al., 2015), with a 0.2 Hz precision of less than $\pm 0.001$ ppm, $\pm 0.0030\%$, $\pm 0.015$ ppm and
$\pm 0.050$ ppm for [CH$_4$], [H$_2$O], [CO] and [CO$_2$], respectively (Picarro, Inc., 2021). The Picarro G2401 streams data directly to the logging computer; this simultaneous HPR logging eliminates time offset issues. Any lag time between the System B sampling cell and the Picarro G2401 was measured and corrected for (typically a few seconds).

As Figaro sensors naturally operate in wet conditions, a dew-point generator (LI-610, LI-COR, Inc., Lincoln, Nebraska, USA) was employed during all System B testing. In addition, mass-flow controllers (Bronkhorst High-Tech B.V., AK Ruurlo,
Netherlands) were utilised, to produce various gas blends at a constant net 1 dm$^3$ min$^{-1}$ flow rate. This is essential to maintain a consistent Figaro cooling effect inside the cell.

## 3. Sensor characterisation

### 3.1 Sensor gas response

Here we describe the general sampling strategy, used to derive [CH$_4$]. According to the Figaro sensor characterisation strategy
of van den Bossche et al. (2017) and Jørgensen et al. (2020), [CH$_4$] can be derived by comparing measured resistance to a baseline reference resistance ($R_b$) measured with a standard gas (Eugster and Kling, 2012). If this reference resistance is well-



characterised to account for environmental changes (independent of gas composition), a gas derivation function ($f$) may be used to yield [$CH_4$], as in Eq. (2), where [$CH_4$]$_b$ is the baseline reference [$CH_4$] in standard gas. This function is independent of environmental variables, as they are already incorporated in the reference resistance and thus, cancel out. Therefore, this
ratio is solely a function of gas enhancement.

$$f(([CH_4] - [CH_4]_b), \ldots) = \frac{R}{R_b} \tag{2}$$

The $f$ function may be dependent on various reducing or oxidising gases, though only $CH_4$ is explicitly included here, for simplicity.

## 3.2 Choice of standard reference gas

In order to conduct repeatable testing, a reliable reference gas is first required. This gas must produce a consistent Figaro resistance response. Our initial candidate was gas from a zero-air generator (UHP-300ZA-S, Parker Hannifin Manufacturing Limited, Gateshead, Tyne and Wear, UK); this catalytic oven oxidises hydrocarbons and CO, resulting in a clean air stream containing 0.00 ppm [$CH_4$] and 0.00 ppm [CO], as recorded by the Picarro G2401. This reference gas was initially selected for testing due to enhanced Figaro environmental sensitivity expected in the absence of all reducing gases (Bastviken et al.,
2020). Zero-air has also been employed as a reference gas by Jørgensen et al. (2020).

But before this zero-air source could be used as a standard gas in subsequent testing, it was important to verify that we could predict the resistance change under a [$CH_4$] transition from 0 ppm to 2 ppm (ambient background), which would be a crucial step in working with zero-air as a standard reference. This test was conducted with various gas samples containing the same 2 ppm [$CH_4$] from different sources, which were sampled with five sensors (LSCE001, LSCE003, LSCE005, LSCE007 and
LSCE009) in System B. First, a cylinder containing 5% [$CH_4$] in argon (P5-Gas ECD, Linde Gas AG, Höllriegelskreuth, Germany) was diluted with 99.996% zero-air generator gas, targeting 2 ppm [$CH_4$]. This was sampled twice. Next, a synthetic air cylinder containing 2 ppm [$CH_4$] (Deuste Gas Solutions GmbH, Schömberg, Germany) was sampled twice. Although this cylinder also contained 5 000 ppm [$CO_2$], this is irrelevant in the context of Figaro resistance response (see Sect. 3.6). All synthetic air cylinders contain a natural balance of nitrogen, oxygen and argon. This was directly followed by sampling two
ambient air sources once: ambient laboratory air from the room surrounding the instruments was sampled for 5 minutes, before finally sampling an ambient target gas cylinder, filled with outdoor air from next to our laboratory some months previous. Ambient is defined here to be any natural air acquired from the surrounding environment.

A dew-point setting of 8° C was applied throughout this test, resulting in (0.970±0.002)% [$H_2O$]. The sensors were allowed to stabilise in response to this [$H_2O$] setting for at least 24 hours directly preceding the test, until there was no noticeable resistance
drift. This stabilisation period is essential, as Figaro sensors exhibit a delayed response to [$H_2O$] changes (see Appendix B).



Results of this 2 ppm [CH$_4$] transition test are presented in Fig. 5. The Picarro G2401 recorded 2 ppm [CH$_4$] for all four gas samples, with consistently low [CO]. However, Figaro resistance decrease varied considerably (see Table 1 for fractional decrease values). Resistance drop (compared to zero-air generator gas) when sampling both ambient target gas and ambient laboratory air was smaller (on average 4% for all five sensors) than when sampling synthetic air and diluted 5% [CH$_4$] (on average 12% for all five sensors), although there was considerable variability between the different sensors (see Table 1). This suggests that there may be one (or many) additional species in ambient air, causing an unexpectedly high Figaro resistance drop. Such a substance may be absent in synthetic air and combusted by the zero-air generator. However, identifying such species remains a challenge (see Sect. 5.2 for discussion), with us unable to identify any obvious alternative ambient reducing candidates from previous Figaro testing work. Moreover, the consistent resistance drop for both synthetic 2 ppm [CH$_4$] and zero-air blended with 0.004% of 5% [CH$_4$], suggests that synthetic 2 ppm [CH$_4$] contains no reducing contaminants.

| methane source | LSCE001 | LSCE003 | LSCE005 | LSCE007 | LSCE009 |
|---|---|---|---|---|---|
| diluted 5% methane | −3% | −4% | −3% | −3% | −3% |
| synthetic air | −4% | −5% | −3% | −3% | −3% |
| ambient laboratory air | −19% | −23% | −7% | −8% | −4% |
| ambient target gas | −19% | −23% | −6% | −8% | −4% |

**Table 1: Fractional Figaro resistance decrease in response to different sources of 2 ppm methane mole fraction, compared to zero-air generator gas. The final 120 s of each 2 ppm sampling period was used to derive these values. A zero-air reference resistance was derived by taking the average of all 120 s zero-air averages, preceding a 2 ppm transition.**

Although this test infers the presence of an interfering substance in ambient natural air (both target gas and laboratory air), it is important to verify that the zero-air generator is not itself a source of such components. It is also useful to test that different synthetic air cylinders (filled at different times) from the same supplier (Deuste Gas Solutions GmbH) behave in the same way, compared to zero-air generator gas. System B was used to sample a synthetic 50 ppm [CH$_4$] cylinder filled in 2019 (old), a synthetic 50 ppm [CH$_4$] cylinder filled in 2021 (new), a synthetic zero-air cylinder filled in 2014 (old) and a synthetic zero-air cylinder filled in 2021 (old), which were all sampled twice. Four sensors were tested (LSCE002, LSCE004, LSCE006 and LSCE008) at a fixed dew point, resulting in (0.652±0.010)% [H$_2$O] for this test. A sufficient [H$_2$O] stabilisation period preceded this test.

Fig. 6 shows Figaro and HPR observations from this test. The two synthetic 50 ppm [CH$_4$] cylinders produced identical resistance decreases, compared to gas from the zero-air generator, when filled two years apart. This suggests that the quality of synthetic 50 ppm [CH$_4$] is consistent and that CH$_4$ is the dominant reducing species in these cylinders. The second part of the test shows that synthetic zero-air has a negligible effect on Figaro resistance, compared to gas from the zero-air generator. Though synthetic zero-air causes a small resistance variability (particularly for LSCE006; see Fig. 6), this is insignificant in the context of the values presented in Table 1, for different 2 ppm [CH$_4$] sources. This consistency in zero-air resistance response suggests that the zero-air generator successfully burns Figaro-sensitive species. This supports the conclusions derived



from Fig. 5 that there may be an additional reducing substance in natural air, otherwise absent in zero-air from multiple sources (both synthetic and from the zero-air generator).

To summarise, these two tests infer that zero-air (either synthetic or from a generator) is an unsuitable standard reference gas. Figaro resistance is abnormally high in zero-air, due to the possible absence of (non-$CH_4$) interfering reducing species otherwise present in ambient air. The fact that the resistance drop in ambient laboratory air was almost identical to the resistance drop in ambient target gas (filled some months previous), suggests that any unidentified background reducing species are stable, with a long lifetime. Elsewhere, Jørgensen et al. (2020) found that a laboratory calibration conducted with zero-air could not be applied to ambient air sampling, which required its own calibration (attributing this to power supply issues). van den Bossche et al. (2017) also found that applying a calibration made in synthetic air to ambient air resulted in larger sensor disparity, compared to an HPR. They attributed this to ±2% oxygen mole fraction ([$O_2$]) variability in their synthetic air source (van den Bossche et al., 2017), however our oxygen test (see Sect. 3.7) shows that this is unlikely and an interfering species was probably responsible. Yet, during our tests, we were unable to identify such interfering species from our HPR and there are no obvious reducing candidates in ambient air (see Sect. 5.2 for discussion). The oxidising capacity of air is unlikely to vary, as surface [$O_2$] is near constant. Furthermore, Collier-Oxandale et al. (2018) observed no ozone sensitivity for the similar Figaro TGS 2600 sensor.

Therefore, to incorporate this natural background effect into any subsequent models or analysis, natural ambient air should be employed as a standard gas instead of zero-air, assuming that the ambient air background composition remains consistent in various characterisation tests. Although natural air contains both $CH_4$ and CO, their variability is typically small, when not in the close vicinity of emission sources. Hence all subsequent testing assumes an ambient 2 ppm [$CH_4$] background.

**3.3 Reference resistance characterisation**

Having selected natural ambient air as a standard gas, the next step is to characterise a standard 2 ppm [$CH_4$] baseline reference resistance ($R_2$) in response to environmental variables, which dominate Figaro performance (Eugster and Kling, 2012, Collier-Oxandale et al., 2019, Rivera Martinez et al., 2021). The most important environmental factors (discussed in Sect. 1) are temperature and [$H_2O$] (Eugster et al., 2020), which were characterised using a large environmental chamber (UD500 C, Angelantoni Test Technologies Srl, Massa Martana, Italy) to simultaneously test five System A loggers. The chamber was slowly replenished (at less than $0.5$ dm$^3$ min$^{-1}$), to avoid Figaro waste gas accumulation, which is slightly enhanced in CO due to some incomplete $CH_4$ surface combustion (Glöckler et al., 2020). Rather than using a solar panel, each System A battery was connected directly to a battery charger, to maintain a stable supply voltage. System A data was remotely accessed by connecting the cellular board inside each enclosure to an antenna outside the chamber. The Picarro G2401 HPR continuously sampled inside the chamber during testing. All System A data was interpolated to the shorter Picarro G2401 timestamp.





Chamber testing was conducted across a temperature and [H₂O] range expected in the field, as suggested elsewhere (Barsan
et al., 2007), to optimise time resources with limited chamber access. [H₂O] of 0.4%, 0.7%, 1.0%, 1.4% and 1.9% were
targeted, by adjusting relative humidity inside the chamber, according to the temperature setting. Following each new [H₂O]
change, the chamber was first given one 7-hour adjustment period, to augment [H₂O] stabilisation, as required in response to
sharp [H₂O] changes (see Appendix B). Next, at least four different temperature settings were sampled at each [H₂O] level in
4-hour intervals (including time for each temperature ramp). Finally, temperature was varied in 8-hour sampling intervals at
the fixed [H₂O] level. Then the entire process was repeated at a different targeted [H₂O].

Chamber observations from each System A logger are presented in Fig. 7. Corresponding HPR measurements, SHT85 $T_A$
measurements and derived SHT85 [H₂O] values are also shown in Fig. 7. [H₂O] averages were derived using SHT85 $T_A$ and
relative humidity measurements from inside each System A enclosure, where saturation vapour pressure was derived using
Teten's equation, given by Murray (1967), and pressure was assumed to be $10^5$ Pa. There was a data transmission gap between
17:14 UTC on 7 December 2021 and 00:46 UTC on 8 December 2021.

The 4-hour intervals presented in Fig. 7 are of insufficient duration for Figaro stabilisation. Despite our efforts to maintain a
fixed [H₂O] level during temperature variations, there was a sharp [H₂O] change at each temperature transition with regular
fluctuations in [H₂O] during each sampling period (see Fig. 7). Thus 30-minute averages were taken towards the end of each
8-hour sampling period, for optimal sensor stabilisation, ranging between 10 kΩ and 47 kΩ for the five sensors. Averages
from 4-hour intervals were discarded, thus conveniently avoiding the data transmission gap. These chamber averages showed
that [H₂O] is the dominant factor influencing $R_2$, as observed in other work (Bastviken et al., 2020, Rivera Martinez et al.,
2021), exhibiting a linearly decreasing relationship. Therefore, Eq. (3) was proposed to model $R_2$ in the environmental
chamber. This equation is analogous to Eq. (2), where $R_2$ is specifically used in place of a general $R_b$ value.

$$R_2 = A \cdot \left( 1 - \left( [\mathrm{H_2O}] \cdot \left( B - (T_A \cdot C) \right) \right) - (T_A \cdot D) \right) \tag{3}$$

$A$ is a baseline reference resistance offset in kΩ, $B$ is a water correction coefficient in %$^{-1}$, $C$ is a temperature-water correction
coefficient in kK$^{-1}$ %$^{-1}$ and $D$ is temperature correction coefficient in kK$^{-1}$, where "%" is a percentage water mole fraction.
[H₂O] here represents a derived value from the SHT85 inside each System A enclosure.

A non-linear regression was applied between $R_2$, $T_A$ and [H₂O] from all 30-minute averages from the 8-hour sampling periods
for each sensor. Model results are presented in Fig. 8 and corresponding model coefficients in Table 2. As Eq. (3) contains
four free parameters, with a limited number of sampling data points, we evaluated the suitability of parameterisation. An
Akaike information criterion (AIC) and Bayesian information criterion (BIC) score was derived for simplified variations of
Eq. (3), with one, two and three free parameters. Results are presented in Table 3, where a lower AIC and BIC score represents
a better compromise, providing a good model fit without over-parameterisation. The results in Table 3 is show that, on average,





the full version of Eq. (3) with four free parameters results in the lowest AIC and BIC score, supporting our four-parameter

approach.

| sensor | $A$ (k$\Omega$) | $B$ (%$^{-1}$) | $C$ (kK$^{-1}$ %$^{-1}$) | $D$ (kK$^{-1}$) | $R^2$ | RMSE (k$\Omega$) | $R_2$ at 25° C $T_A$ and 1% [H$_2$O] (k$\Omega$) | RMSE as a fraction of $R_2$ at 25° C $T_A$ and 1% [H$_2$O] (%) |
|---|---|---|---|---|---|---|---|---|
| LSCE001 | 30.7 | 0.389 | 0.924 | 1.46 | 0.961 | ±0.39 | 13.9 | ±2.8 |
| LSCE003 | 29.5 | 0.377 | 0.833 | 1.24 | 0.959 | ±0.43 | 14.8 | ±2.9 |
| LSCE005 | 75.8 | 0.419 | 1.135 | 2.10 | 0.980 | ±0.52 | 22.2 | ±2.4 |
| LSCE007 | 44.7 | 0.317 | 0.680 | 1.45 | 0.970 | ±0.51 | 20.3 | ±2.5 |
| LSCE009 | 164.3 | 0.443 | 1.295 | 2.40 | 0.974 | ±0.99 | 37.4 | ±2.6 |

Table 2: Eq. (3) model parameters for five System A enclosures, derived from 30-minute averages (of 8-hour testing windows), whilst sampling natural ambient air in the environmental chamber. The $R^2$ and RMSE is given for each model fit and the RMSE is given as a fraction of $R_2$ at 25° C $T_A$ and 1% [H$_2$O], for each sensor.

| equation ($R_2$ =) | $A \cdot (1 - ([H_2O] \cdot (B - (T_A \cdot C))) - (T_A \cdot D))$ | | $A \cdot (1 - ([H_2O] \cdot B) - (T_A \cdot D))$ | | $A \cdot (1 - ([H_2O] \cdot B))$ | | $A \cdot (1)$ | |
|---|---|---|---|---|---|---|---|---|
| test | AIC | BIC | AIC | BIC | AIC | BIC | AIC | BIC |
| LSCE001 | 424 | 431 | 424 | 429 | 423 | 427 | 509 | 512 |
| LSCE003 | 429 | 436 | 428 | 434 | 427 | 431 | 513 | 515 |
| LSCE005 | 440 | 447 | 447 | 452 | 454 | 458 | 544 | 546 |
| LSCE007 | 439 | 445 | 438 | 443 | 439 | 443 | 531 | 534 |
| LSCE009 | 476 | 482 | 487 | 493 | 497 | 501 | 571 | 574 |
| average | 441±18 | 448±18 | 445±23 | 450±23 | 448±27 | 452±27 | 534±23 | 536±23 |

Table 3: AIC and BIC scores for simplified variations of the Eq. (3) model for five System A enclosures, derived from 30-minute

averages (of 8-hour testing windows), whilst sampling natural ambient air in the environmental chamber.

Having selected the four-parameter model given by Eq. (3), the RMSE $R_2$ when modelling environmental chamber sampling was derived and is provided in Table 2, spanning between ±0.4 k$\Omega$ and ±1.0 k$\Omega$. This represents less than ±3% fractional uncertainty at 25° C $T_A$ and 1% [H$_2$O], for all five sensors. This low model error suggests that Eq. (3) provides good temperature and [H$_2$O] constraint to $R_2$. Furthermore, a coefficient of determination (R$^2$) of 0.97±0.01 for the five model fits illustrates the

suitability of Eq. (3) in characterising $R_2$, with respect to environmental conditions (see Table 2 for values).

### 3.4 Methane characterisation

In order to derive a Figaro CH$_4$ response function, the effect of adding CH$_4$ to standard gas (natural ambient air) was characterised by testing five Figaro sensors (LSCE001, LSCE003, LSCE005, LSCE007 and LSCE009), using System B. Ambient laboratory air was blended with gas from a cylinder containing 5% [CH$_4$] in argon (P5-Gas ECD, Linde Gas AG), in

15-minute intervals from 2 ppm (ambient laboratory air) up to a 1 000 ppm target [CH$_4$]. This 1 000 ppm gas blend has a argon mole fraction enhancement of 145% and an oxygen and nitrogen mole fraction diminution in of 1.44%, compared to natural ambient air. This 1 000 ppm level represents a realistic upper limit on typical [CH$_4$] enhancements expected in the vicinity of



most methane sources, such as large leaks from oil and gas extraction infrastructure. This high upper [CH$_4$] limit also facilitates better sensor characterisation over an extended range. Following at least 1 hour of ambient laboratory air sampling, [CH$_4$] was

gradually raised up to its maximum level and then lowered, step-wise, in three cycles. After each cycle, ambient laboratory air was sampled for 1 hour to provide an $R_2$ reference. This approach is similar to that of Jørgensen et al. (2020), who instead transitioned back to their standard gas following each gas enhancement. Throughout our test, an 8° C dew-point setting was applied, which was sampled from at least 24 hours in advance to facilitate the necessary water stabilisation (see Appendix B).

Full Figaro resistance results are presented in Fig. 9. Fig. 10 provides an example of a single [CH$_4$] transition for LSCE001,

where the final 2 minutes of 15-minute sampling intervals are highlighted. This shows that the final 2 minutes is a suitable representation of stable Figaro resistance, thanks to efficient cell flushing, unlike a long cell residence time observed in other work (Rivera Martinez et al., 2022). Fig. 10 also shows that there is little noise in System B Figaro response. Therefore a 2-minute average was derived at the end of each 15-minute sampling period (highlighted in Fig. 9). A specific $R_2$ reference baseline was then derived for this test by fitting a second order polynomial to the final 15 minutes of each 1-hour standard

(ambient laboratory air) sampling period, except the first period, where 45 minutes of sampling was instead used (see Fig. 9). $R_2$ was not derived from Eq. (3) in this test as, Eq. (3) is only valid in System A. This dynamic $R_2$ incorporates any reference resistance variability during the test, which may occur due to small environmental changes. In any case, temperature and [H$_2$O] both remained stable: [H$_2$O] was on average (1.002±0.001)% during $R_2$ sampling periods, according to the Picarro G2401 HPR, and System B temperature was on average (34.2±0.2)° C, according to the SHT85 inside the System B cell.

For each 2-minute Figaro resistance average, corresponding Picarro G2401 [CH$_4$] averages were derived. Wet [CH$_4$] is used here and throughout this manuscript, to minimise errors associated with the internal Picarro G2401 water correction, especially at higher [CH$_4$], where spectral overlap becomes more prominent and [H$_2$O] measurements become less reliable. For [CH$_4$] of over 100 ppm, [CH$_4$] was instead derived from the mass-flow controller setting, as the Picarro G2401 is less precise at high [CH$_4$]. Water was then reintroduced into these dry [CH$_4$] estimates. The ratio between each measured resistance average and

its corresponding polynomial $R_2$ estimate was then deduced and plotted against its respective [CH$_4$] value in Fig. 11.

Fig. 11 suggests that resistance ratio follows a power law decay behaviour, whereby resistance ratio slowly tends towards zero, as [CH$_4$] enhancement (above the 2 ppm standard) tends to infinity. However, a simple power law fit cannot be used here: when mole fraction enhancement is equal to zero (*i.e.* when [CH$_4$] is equal to the 2 ppm standard), the resistance ratio must be equal to unity (*i.e.* $R_2$ must equal $R$). Therefore, Eq. (4) is proposed, where one is added to the CH$_4$ gas term to satisfy this

requirement.

$$R = R_2(T_A, [H_2O]) \cdot \left( 1 + \left( \frac{[CH_4] - 2\,\text{ppm}}{a} \right) \right)^{-\alpha} \cdot \prod_g \left( 1 + \left( \frac{[M_g] - [M]_{0_g}}{c_g} \right) \right)^{-\gamma_g} \qquad (4)$$





$a$ is the characteristic methane mole fraction and $\alpha$ is the methane power. Other reducing gases ($g$) may be included in Eq. (4) depending on sampling conditions, where [$M$] is the mole fraction of $g$, [$M$]$_0$ is the standard mole fraction of $g$ (in ambient air), $c$ is the characteristic mole fraction of $g$ and $\gamma$ is the power of $g$. Eq. (4) is a general equation which allows any potential reducing gases to be incorporated in Figaro resistance response. However, for a more specific case when [$M$] is equal to [$M$]$_0$, as in standard gas, these multiplicative terms tend to unity and can be ignored from Eq. (4), thus simplifying to Eq. (5).

$$R \approx R_2(T_A, [H_2O]) \cdot \left(1 + \left(\frac{[CH_4] - 2\,ppm}{a}\right)\right)^{-\alpha} \tag{5}$$

Thus, rather than deriving $c$ and $\gamma$ for each potential reducing gas, Eq. (5) only focuses on a single variable gas ($CH_4$, in this case) responsible for most resistance variability.

This model fits System B measurements of resistance ratio (*i.e.* measured resistance averages divided by their corresponding polynomial $R_2$ estimates) from the $CH_4$ characterisation test very well (see Table 4 for $a$ and $\alpha$ for the five tested sensors), resulting in an RMSE resistance ratio of no more than $\pm1\%$ $\Omega\,\Omega^{-1}$ and a $R^2$ of 0.9993$\pm$0.0005, for the five sensors. This means that over a 1 000 ppm [$CH_4$] range, the ratio between measured Figaro resistance and standard reference resistance can be predicted to within $\pm1\%$, thus allowing [$CH_4$] estimates to be derived by comparing measured resistance to $R_2$. Eq. (5) was also inverted to make [$CH_4$] the subject. Using the same original fitting parameters provided in Table 4, this revealed an inverted [$CH_4$] RMSE of no more than $\pm31$ ppm for the model fit, over the full 1 000 ppm range (see Table 4 for individual values). A [$CH_4$] threshold reduced this uncertainty further, as [$CH_4$] is more accurate at lower [$CH_4$], where there were more data points. Taking [$CH_4$] values of 28 ppm and lower (nine targeted [$CH_4$] levels) and using the same fitting parameters from the extended [$CH_4$] range, resulted in a reduced inverted [$CH_4$] RMSE uncertainty of no more than $\pm0.85$ ppm. Though it is possible to derive better fitting parameters in this reduced [$CH_4$] range, the extended [$CH_4$] range permits better characterisation of the natural power decay behaviour. Furthermore, characterising only small [$CH_4$] enhancements limits the model to such circumstances; this may be desirable in cases where there is certainty that sampled [$CH_4$] enhancements will remain low.

| sensor | $a$ (ppm) | $\alpha$ | $R^2$ | RMSE ($\Omega\,\Omega^{-1}$) | inverted RMSE (ppm) | inverted RMSE with 28 ppm [$CH_4$] threshold (ppm) | resistance ratio at 3 ppm [$CH_4$] ($\Omega\,\Omega^{-1}$) | resistance ratio at 50 ppm [$CH_4$] ($\Omega\,\Omega^{-1}$) |
|---|---|---|---|---|---|---|---|---|
| LSCE001 | 26.3 | 0.368 | 0.9997 | $\pm$0.0038 | $\pm$12 | $\pm$0.37 | 0.986 | 0.683 |
| LSCE003 | 23.2 | 0.357 | 0.9997 | $\pm$0.0041 | $\pm$16 | $\pm$0.41 | 0.985 | 0.670 |
| LSCE005 | 30.2 | 0.461 | 0.9993 | $\pm$0.0068 | $\pm$15 | $\pm$0.68 | 0.985 | 0.645 |
| LSCE007 | 31.3 | 0.439 | 0.9993 | $\pm$0.0065 | $\pm$13 | $\pm$0.69 | 0.986 | 0.665 |
| LSCE009 | 24.7 | 0.502 | 0.9986 | $\pm$0.0099 | $\pm$31 | $\pm$0.85 | 0.980 | 0.582 |

**Table 4: Eq. (5) methane model parameters for five Figaro sensors, with the $R^2$ and RMSE for each model fit. Inverted methane mole fraction RMSE values are also given over the full 1 000 ppm range and with a 28 ppm threshold. The expected ratio between measured resistance and $R_2$ is also provided for a 1 ppm and 48 ppm [$CH_4$] enhancement above the 2 ppm background.**





Although there is a good $CH_4$ model fit for the extended [$CH_4$] range, in practice, [$CH_4$] can only be derived from the ratio between measured resistance and $R_2$. The resistance ratio for a 1 ppm enhancement above the background (to 3 ppm [$CH_4$]) would be between $0.980\,\Omega\,\Omega^{-1}$ and $0.986\,\Omega\,\Omega^{-1}$ for the five tested sensors, while resistance ratio for a 48 ppm enhancement above the background (to 50 ppm [$CH_4$]) would be between $0.582\,\Omega\,\Omega^{-1}$ and $0.683\,\Omega\,\Omega^{-1}$ (see Table 4 for individual values).

This makes small [$CH_4$] enhancements difficult to detect; a transition from 2 ppm to 3 ppm [$CH_4$] results in a resistance drop of as little as 1%. Thus, [$CH_4$] estimation using Eq. (5) requires good modelled $R_2$ estimates (from sect. 3.3), in order to derive a reliable resistance ratio.

**3.5 Carbon monoxide influence**

[CO] can vary in natural ambient air depending on nearby pollution (*e.g.* petrol and diesel cars), but is typically of the order

of $10^{-1}$ ppm. As CO is a potent reducing gas, the importance of CO variations within standard ambient air was tested with four sensors (LSCE002, LSCE004, LSCE006 and LSCE008) in System B. Figaro resistance at 0.1 ppm [CO] was compared to a 0.0 ppm [CO] standard baseline reference (with only CO removed). An ambient target gas cylinder, filled with outside air (2 ppm [$CH_4$] and 0.15 ppm [CO]) was split into two gas streams: one stream was directly from the cylinder and the other stream passed through a chemical CO scrubber (Sofnocat 514, Molecular Products, Limited, Harlow, Essex, UK). The 0.0 ppm

[CO] reference was first sampled for at least 1 hour. Then, 0.1 ppm [CO] was sampled in four 15-minute intervals. Each 0.1 ppm interval was followed by 15 minutes sampling the 0.0 ppm [CO] reference. A fixed 8° C dew point setting was applied and a sufficient [$H_2O$] stabilisation period preceded this test.

Figaro resistances and corresponding HPR measurements are presented in Fig. 12. [$CH_4$] remained fixed at 2 ppm throughout this test, allowing us to assess the independent influence of CO on Figaro resistance, in standard gas. A 5-minute average was

taken from the end of each 15-minute 0.1 ppm [CO] sampling period (highlighted in Fig. 12). A baseline reference was then derived by fitting a second order polynomial to the final 5 minutes of each 15-minute reference (0.0 ppm [CO]) sampling period, except the first period where 45 minutes was used (see Fig. 12). [$H_2O$] was on average $(0.983\pm0.001)$% during these reference sampling periods, according to the Picarro G2401, and System B temperature was on average $(31.3\pm0.1)°$ C, according to the SHT85 sensor inside the cell.

The resistance ratio between each 5-minute 0.1 ppm [CO] average and its corresponding modelled reference (0.0 ppm [CO]) resistance was derived. Four individual resistance ratios were acquired and then averaged for each sensor: $(0.9922\pm0.0006)\,\Omega\,\Omega^{-1}$ for LSCE002, $(0.9936\pm0.0006)\,\Omega\,\Omega^{-1}$ for LSCE004, $(0.9960\pm0.0009)\,\Omega\,\Omega^{-1}$ for LSCE006 and $(0.9950\pm0.0005)\,\Omega\,\Omega^{-1}$ for LSCE008. Thus, a standard gas transition from 0.0 ppm to 0.1 ppm [CO] results in less than 1% resistance decrease. This low CO sensitivity is due to the incorporation of an internal CO filter. This small CO resistance effect

could become important in the context of small [$CH_4$] variations accompanied by an incredibly stable $R_2$ baseline, allowing miniscule resistance variations can be observed. However, in typical applications, less than 1% resistance change will not be





an important factor and thus CO can usually be excluded from Eq. (4). Furthermore, gas sensitivity declines with increasing mole fraction (*i.e.* a [CO] transition from 0.1 ppm to 0.2 ppm will result in an even smaller resistance decrease).

**3.6 Carbon dioxide response**

Figaro sensors naturally respond to reducing gases. As $CO_2$ is the most oxidised gaseous form of carbon (with no reducing potential), it is not expected to influence Figaro resistance. To verify a null $CO_2$ effect, two synthetic air cylinders (Deuste Gas Solutions GmbH) containing 5 000 ppm [$CO_2$] and 1 000 ppm [$CO_2$] were sampled, using System B. Both cylinders contained similar ambient quantities of $CH_4$ and CO. After sampling gas from the zero-air generator, each cylinder was sampled for two short intervals, before returning to zero-air generator gas. Then an ambient target gas cylinder, filled with outside air, was
sampled. Four sensors were tested (LSCE002, LSCE004, LSCE006 and LSCE008) at a fixed dew point, resulting in [$H_2O$] of (0.649±0.006)% for this test. A sufficient water stabilisation period preceded this test.

Figaro sampling results are presented in Fig. 13, alongside corresponding HPR measurements. Fig. 13 shows that both synthetic air sources result in the same Figaro resistance decrease. This consistent decrease is principally due to the similar [$CH_4$] content of both cylinders. Meanwhile ambient target gas results in a much larger resistance decrease, as observed in
Sect. 3.2. Therefore, $CO_2$ can rightly be eliminated as a species of concern when dealing with Figaro resistance output.

**3.7 Oxygen response**

Oxygen naturally forms 20.95% of dry air, at sea level. As an oxidising gas, increasing [$O_2$] should elevate Figaro resistance, in contrast to the opposite effect of reducing gases, such as $CH_4$. To verify this behaviour and to quantify the importance of [$O_2$] variability, zero-air generator gas was diluted with nitrogen gas (99.999%, Air Products SAS, Saint Quentin Fallavier,
France), using System B. Following at least 1 hour of zero-air sampling, [$O_2$] was gradually depleted to half its ambient atmospheric background, stepwise, in 15-minute intervals in three cycles. Each cycle was concluded with a 45-minute period of sampling zero-air generator gas. Five Figaro sensors were tested (LSCE002, LSCE004, LSCE006, LSCE008 and LSCE010) at an 8° C dew point. A sufficient water stabilisation period preceded this test.

2-minute average resistances were taken from the end of each 15-minute sampling period (see Fig. 14). Corresponding wet
[$O_2$] estimates were derived for each resistance average, using the mass-flow controller setting and [$H_2O$]. An [$H_2O$] value of (1.008±0.002)% was derived from the Picarro G2401 during 2-minute averages at the maximum [$O_2$] level (other HPR measurements could not be used due to peak broadening effects at lower [$O_2$]). Average Figaro resistance is plotted against [$O_2$] in Fig. 14. Decreasing [$O_2$] leads to a reduced Figaro resistance, in agreement with other SMO sensors (Yang et al., 2020). This behaviour is expected for Figaro sensors (van den Bossche et al., 2017, Glöckler et al., 2020), as desorbing oxygen from
the SMO surface releases electrons into the conduction band. For the five tested Figaro sensors, a 1.8% [$O_2$] drop results in a (0.8±0.1)% Figaro resistance decrease. Furthermore, inferring a linear fit between the highest two [$O_2$] points reveals a





(0.0021±0.0003)% Figaro resistance decrease corresponding to a [$O_2$] decrease of 0.001% (10 ppm), typical of natural ambient [$O_2$] variability. This small effect means that oxygen can be ignored from most Figaro characterisation work, as near-surface changes in ambient [$O_2$] are negligible. This test also shows that Figaro sensors are insensitive to small changes in oxygen partial pressure (which is directly proportional to [$O_2$], at fixed atmospheric pressure). Oxygen partial pressure is also directly proportional to net atmospheric pressure (at fixed [$O_2$]). Thus, we can infer from this test that Figaro resistance response is insensitive to small changes in net atmospheric pressure.

## 4. Field testing

### 4.1 Field deployment

Here we discuss Figaro autonomous field testing. All ten System A loggers were deployed at the SUEZ Amailloux landfill site in the west of Metropolitan France (46.7568° N, 0.3547° E). A landfill site served as an ideal initial field testing location, as it is a large area emission source producing methane throughout the year, with occasional [$CH_4$] enhancements above the background of the order of $10^1$ ppm. SUEZ Amailloux landfill topography gradually evolves over time, as new cells are opened, filled and then covered over with soil and geomembrane. The site features biogas collection infrastructure, in common with other European landfills (Daugela et al., 2020). The location of the ten System A loggers is provided in Fig. 15, with an example of field installation shown in Fig. 3. The loggers were typically positioned on covered soil, away from any direct point emission sources, except for LSCE003, which was placed near a leaking vent. Three loggers were moved from an "old" to "new" location, due to site evolution: LSCE001 was moved between July and November 2021; LSCE010 was moved between February and March 2022; LSCE009 was moved on 28 April 2021.

As [$CH_4$] response (Sect. 3.4) and $R_2$ (Sect. 3.3) characterisation tests have been performed on five sensors (LSCE001, LSCE003, LSCE005, LSCE007 and LSCE009), these five System A loggers will be the focus of subsequent analysis. These sensors sampled in the field between 20 March 2021 and 16 November 2021 (period 1) and then between 22 December 2021 and 27 March 2022 (period 2). Sensor testing was performed in-between these two sampling periods. LSCE005 stopped transmitting data on 19 October 2021. Other minor data gaps occurred due to data transmission issues.

### 4.2 Reference resistance modelling

For the five selected Figaro sensors, $R_2$ was modelled for all field sampling, using Eq. (3). The ratio between measured resistance and $R_2$ may then subsequently be used to derive [$CH_4$], following Eq. (5). The $R_2$ model used, as input, raw measured $T_A$ and [$H_2O$] derived from the SHT85 inside each System A enclosure. [$H_2O$] was derived using the same procedure outlined in Sect. 3.3 (Murray, 1967). Modelled $R_2$ for the five System A loggers is presented in Fig. 16 for period 1 and in Fig. 17 for period 2.





Fig. 16 and Fig. 17 show that the Eq. (3) $R_2$ model can replicate some features of measured resistance, due to the incorporation of water and temperature effects. The Person correlation coefficient ($P$) between measured resistance and $R_2$ (given in Table 6) is greater than half for all bar one sensor (LSCE003), during both period 1 and period 2. Poor correlation for LSCE003 is hardly surprising, considering its placement near to a leaking vent. Yet for all five sensors there is a general disparity between modelled $R_2$ and measured resistance, which outweighs any correlation, based on average resistance ratios for both periods, provided in Table 6. For reference, a ratio between measured resistance and $R_2$ of one corresponds to [CH$_4$] of 2 ppm (standard air). Thus, Table 6 values should thus be close to one, or slightly less than one if generally sampling [CH$_4$] enhancements, as expected for LSCE003 which is near a methane leak. A ratio more than one (*i.e.* when $R_2$ is less than measured resistance) corresponds to [CH$_4$] below 2 ppm, which is impossible in the absence of a potent CH$_4$ sink.

| sensor | period 1 resistance ratio ($\Omega\,\Omega^{-1}$) | period 1 $P$ | period 2 resistance ratio ($\Omega\,\Omega^{-1}$) | period 2 $P$ |
|---|---|---|---|---|
| LSCE001 | 1.46±0.14 | 0.663 | 1.06±0.11 | 0.733 |
| LSCE003 | 1.20±0.18 | 0.417 | 0.96±0.13 | 0.107 |
| LSCE005 | 1.35±0.11 | 0.822 | 0.89±0.05 | 0.892 |
| LSCE007 | 1.78±0.15 | 0.678 | 1.07±0.05 | 0.874 |
| LSCE009 | 1.08±0.09 | 0.772 | 0.85±0.03 | 0.924 |

**Table 6: The average ratio and $P$ between System A measured resistance and standard 2 ppm [CH$_4$] Figaro reference resistance, while sampling at the SUEZ Amailloux landfill site during period 1 and period 2. Standard deviation uncertainties for resistance ratios are given.**

Table 6 averages suggest that Eq. (3) $R_2$ model performance is unsatisfactory for the ultimate purpose of estimating [CH$_4$], where an enhancement above the background of 1 ppm [CH$_4$] can correspond to a resistance drop of as little as 1%. Fig. 16 shows that during period 1, measured Figaro resistance was larger than $R_2$ (a ratio greater than one) for all five sensors most of the time, except for some overlap for LSCE009 up to June 2021. Resistance disparity was particularly stark for LSCE007, with an average period 1 resistance enhancement of +(78±15)%, compared to $R_2$. Conversely, for period 2, Fig. 17 shows that resistance ratios decreased for all five sensors and were closer to one (see Table 6), resulting in a generally better $R_2$ agreement. However, Fig. 17 shows no period 2 improvement in capturing the nuances of daily temperature and [H$_2$O] variations. For LSCE005 and LSCE009, the period 2 resistance ratio was less than one (within the uncertainty range), which would imply consistently enhanced [CH$_4$] above 2 ppm, otherwise absent during period 1 (unlikely).

The reproduction of an $R_2$ baseline, that can well-incorporate environmental variability, is essential to model [CH$_4$] enhancements above the 2 ppm standard background, using Eq. (5). Based on model $R_2$ and resistance measurements presented in Fig. 16 and Fig. 17, [CH$_4$] cannot be derived here in this way. There may be other factors causing resistance disparity, which must be first addressed, before this sensor can be used to estimate parts-per-million level [CH$_4$] enhancements in future, which we discuss in Sect. 5.1.





## 5. Discussion

### 5.1 Field reference resistance disparity

In this section we attempt to understand the cause of poor agreement between $R_2$ (modelled from temperature and [$H_2O$]) and measured resistance, as presented in Sect. 4.2, and the reasons why reference resistance disparity was different before sensor testing (period 1) compared to after sensor testing (period 2). From Sect. 3.3, the Eq. (3) model yielded excellent $R_2$ agreement during chamber testing (see Fig. 8), with an $R_2$ RMSE below ±1 kΩ for the five tested sensors and an $R^2$ of at least 0.96 (see Table 2). However, modelling $R_2$ in the field was more challenging than in a controlled environment, with disparity between $R_2$ and measured resistance up to the order of $10^1$ kΩ. In addition, this resistance ratio decreased for all five sensors in period 2, though the cause of this change is not clear. Changes in the [$CH_4$] background was unlikely to be responsible, as we also conducted regular onsite and offsite sampling campaigns (not shown), where no excessive abnormalities in general [$CH_4$] variability were observed. Thus, we expect emissions from SUEZ Amailloux to remain at a relatively consistent order of magnitude throughout the year.

One possible cause of poor $R_2$ fitting, was the composition of air during environmental chamber testing. On the one hand, no [$CH_4$] or [CO] irregularities were observed in the chamber by the Picarro G2401 HPR. However, the results presented in Fig. 5 point to the presence of a different reducing species in air, otherwise absent in clean synthetic gas (see Sect. 5.2 for further discussion). It is possible that the composition of these interfering compounds was different in the chamber compared to the landfill site, either through high-temperature chamber degassing or due to the natural ambient composition of the surrounding chamber environment. A cocktail of trace gas species (other than $CH_4$ and $CO_2$) can be emitted from landfill sites, including species such as sulphides, ammonia, alcohols, alkanes, alkenes and aromatics, which vary by many orders of magnitude in different landfill sites (Duan et al., 2020). Yet, the pronounced resistance ratio jump from period 1 to period 2 does not support this hypothesis as the principal cause of resistance disparity. If there were consistently poor $R_2$ model parameters, one would expect field resistance to consistently exceed $R_2$ and not to erroneously decrease in period 2.

Another possibility for poor $R_2$ agreement with measured resistance, is differences in Figaro cooling dynamics in the chamber, compared to the field. van den Bossche et al. (2017) showed that the location of a temperature measurement can be highly influential concerning its application in any correction model. We therefore used the same System A logger in both applications to minimise such effects. Yet, Figaro airflow may still vary depending on exterior System A conditions. In the field, the logging enclosures faced downwards, where lateral winds could influence upwards airflow from the downwards facing fan, due to a vacuum effect. On the other hand, boxes faced sideward in the chamber, with a large chamber fan for air circulation. These two scenarios may have cooled the Figaro sensors inside the System A enclosure differently, such that the temperature gradient between the SHT85 environmental sensor and the Figaro varied, rendering the Eq. (3) $R_2$ model unusable.



Sect. 4 shows that there is an unexplained jump in resistance ratio from period 1 to period 2. Yet, the above discussion suggests that the $R_2$ model may be fundamentally flawed, either due to airflow effects or different levels of other interfering reducing gas species (see Sect. 5.2 for further discussion). Instead of resistance ratio, it may be better to analyse at raw resistance measurements. Maybe, cooler and drier period 2 conditions (largely coinciding with Northern Hemispheric winter) erroneously exaggerated $R_2$. The full $T_A$ and $[H_2O]$ measurement range is presented in Fig. 18 for both periods, for comparison, along with the measured Figaro resistance range. When actual resistance measurements are assessed, there is a large overlap between period 1 and period 2 over the full sampling range. Nevertheless, Fig. 18 shows that measured resistance was significantly lower for all five sensors in period 2, considering the interquartile range, and particularly so for LSCE005 and LSCE007. Yet in view of an equally significant temperature and $[H_2O]$ period 2 decrease, it is possible that these environmental effects may account for the period 2 resistance drop if a better $R_2$ model were used, thus improving $R_2$ agreement with measured resistance.

A final cause of disparity between $R_2$ and measured resistance may be spontaneous variations in the sensor itself, causing the original $R_2$ model parameters to be invalid. However, the fact that resistance ratio decreased for all five sensors in period 2 makes this hypothesis unlikely. Instead, something may have physically altered natural behaviour of multiple sensors during testing, such as the transfer from System A to System B or extreme $[H_2O]$ or temperature conditions. Alternatively, high concentration exposure to certain gases can cause permanent sensor damage, which may have occurred some time between period 1 and period 2. While such effects may have been a contributory factor, the most likely cause of reference resistance disparity from actual resistance measurements (and the change in resistance ratio from period 1 to period 2) is a poor $R_2$ model which did not suitably account for sampling conditions in the field.

## 5.2 Characterisation approach and future improvements

Here we discuss our general Figaro testing approach and compare our methods to other work conducted with the Figaro TGS 2611-E00, along with studies on other Figaro sensor types. In Sect 3.2, we derived $R_2$ using an environmental chamber. Yet according to Eugster et al. (2020), who attempted their own chamber characterisation of the less selective (but more sensitive) Figaro TGS 2600, chamber testing may not be suited for SMO sensors in general. They instead employed a long-term HPR field calibration (Eugster et al., 2020). Field calibration has proved popular for the TGS 2600, where ambient HPR measurements help to optimise model parameters (Eugster and Kling, 2012, Casey et al., 2019, Collier-Oxandale et al., 2019) in conditions with a similar environment and pollutant levels (Collier-Oxandale et al., 2018). An analogous approach can also be applied to ambient laboratory sampling, by simply leaving a sensor to sample in a laboratory alongside an HPR (Martinez Rivera et al., 2021), with an aim for subsequent field deployment (Riddick et al., 2020). Yet, ambient air sensor characterisation can be problematic if various calibration models are required in different conditions, for example in different humidity (Collier-Oxandale et al., 2018) or temperature (Eugster et al., 2020) regimes.





Despite this, the Figaro TGS 2611-E00 has successfully been tested in controlled conditions in the past, for example Cho et al. (2022) set an oven set to three precise temperatures, where [$CH_4$] and relative humidity were externally controlled to fill a 2 dm$^3$ test chamber. Although the application of their calibration model was tested in controlled conditions, it was not HPR

field-tested (Cho et al., 2022). Bastviken et al. (2020) conducted chamber testing at various temperature and humidity settings up to 3.5% [$H_2O$] (humidity was indirectly controlled), where $CH_4$ was injected at each setting. As this calibration was designed to detect high [$CH_4$] in flux chambers, it was not extensively field-tested (Bastviken et al., 2020). van den Bossche et al. (2017) instead tested a Figaro sampling cell in a water bath, for improved temperature regulation. Elsewhere, Jørgensen et al. (2020) conducted laboratory tests at three different relative humidity settings, with no temperature control, assuming constant

laboratory temperature. However, they could not use this test in the field (where zero-air served as a standard gas) and instead employed an HPR field calibration, assuming invariant environmental conditions (Jørgensen et al., 2020).

Yet, a key limitation of ambient air characterisation, is the requirement of an expensive HPR, co-located with each Figaro for a sufficient period of testing time, in order to derive a robust long-term model. Unless readily available, this can negate the central advantage of a cheap SMO sensor. Most of the System A loggers at the SUEZ Amilloux landfill site were isolated and

distant from sources of mains power, typically required by a $CH_4$ HPR. Furthermore, the site is constantly evolving, which is conducive to the deployment of low-cost sensors powered by a solar panel, due to their mobility and ease of remote installation. Thus, we conducted Figaro characterisation in controlled conditions. HPR ambient air testing of ten System A loggers is not logistically feasible. However, it is worth noting that it may be possible to characterise $R_2$ for multiple Figaro sensors using a single HPR, by only selecting sampling during high winds, assuming the wind to sufficiently dilute any nearby methane

emission source. Nevertheless, a [$CH_4$] field characterisation cannot be achieved in this way.

Instead, our controlled calibration approach required the simulation of environmental field conditions. Based on our [$O_2$] test (see Sect. 3.7), atmospheric pressure was dismissed as a key factor effecting the TGS 2611-E00, in agreement with other work (van den Bossche et al., 2017, Rivera Martinez et al., 2021). However, environmental chamber tests revealed a strong [$H_2O$] and temperature resistance response, as observed elsewhere (Bastviken et al., 2020, Rivera Martinez et al., 2021, Cho et al.,

2022). Temperature may also influence electronic measurement circuitry (Ferri et al., 2009). We found [$H_2O$] to dominate resistance, at fixed [$CH_4$]. We accounted for these environmental effects in our calibration approach by deriving a standard $R_2$, following van den Bossche et al. (2017). Whereas van den Bossche et al. (2017) derived logarithmic relationships between environmental parameters and standard resistance, we found linear correlations to be suitable.

Conversely, in many past studies testing the TGS 2600 (Eugster and Kling, 2012, Collier-Oxandale et al., 2018, Eugster et al.,

2020, Riddick et al., 2020), TGS 2602 (Casey et al., 2019, Collier-Oxandale et al., 2019) and TGS 2611-E00 (Bastviken et al., 2020, Jørgensen et al., 2020, Cho et al., 2022), a fixed reference resistance has been used, in contrast to our dynamic $R_2$ approach. Temperature and water effects have then subsequently been incorporated into models, alongside resistance ratio, to yield [$CH_4$] (Collier-Oxandale et al., 2018). Collier-Oxandale et al. (2019) and Casey et al., (2019) used this fixed reference





approach to derive [CH$_4$] (as well as other gas mole fractions) by combining input from various sensors including a TGS 2600 and TGS 2602. Bastviken et al. (2020) used a combination TGS 2611-E00 environmental correction, where water and temperature were first incorporated into a dynamic reference resistance and then subsequently corrected from resistance ratio.

Despite our best efforts, our dynamic $R_2$ model could not replicate field Figaro resistance. One cause may have been a misrepresentative temperature measurement during testing, compared to field sampling (see Sect. 5.1 for specific discussion). In light of this temperature dependence, Eugster et al. (2020) proposed a TGS 2600 heat-loss model using wind speed, temperature and air heat capacities, however, this model could not predict [CH$_4$] any better than their original deterministic model. Elsewhere, Casey et al. (2019) found that low wind speeds adversely affected the performance of both linear and ANN [CH$_4$] models, whose TGS 2600 and TGS 2602 were also inside an enclosure. In light of this potential wind effect, we compared resistance ratio with increasing minute-average wind speed for LSCE007, as measured simultaneously by the LSCE007 System A anemometer (Fig. 19), where wind direction was between 180° and 270° (*i.e.* away from the active landfill). This shows that, there is no correlation between wind speed and resistance ratio, which therefore suggests that our $R_2$ model is not fundamentally influenced by wind speed.

All types of Figaro sensors are clearly affected by water. Yet, when correcting for water effects, some researchers have used relative humidity (Eugster and Kling, 2012, van den Bossche et al., 2017, Jørgensen et al., 2020, Cho et al., 2022), some have used either [H$_2$O] or absolute humidity (Collier-Oxandale et al., 2018, Casey et al., 2019, Collier-Oxandale et al., 2019, Eugster et al., 2020, Rivera Martinez et al., 2021, Rivera Martinez et al., 2022) and some have mixed both in model combinations (Bastviken et al., 2020). As these SMO sensors respond to absolute water content, we chose [H$_2$O] in our model, representing the fraction of water molecules in air. Absolute humidity is a mass fraction, similar to [H$_2$O]. On the other hand, relative humidity represents the proximity to water saturation (dew point), as a function of temperature. Thus [H$_2$O] or absolute humidity typically results in superior model fitting (Bastviken et al., 2020).

Figaro sensors in general require a sufficient warm-up time before testing (Honeycutt et al., 2019, Glöckler et al., 2020, Cho et al., 2022). They also slowly age over time (Eugster et al., 2020, Riddick et al., 2020), resulting in reduced sensitivity (Eugster and Kling, 2012, Collier-Oxandale et al., 2018). Collier-Oxandale et al. (2019) resolved ageing effects by including time as a reference resistance parameter. In principle, ageing can easily be corrected by fitting between calibrations performed at two time points (Eugster and Kling, 2012). While, Riddick et al. (2020) recommend bimonthly calibrations to account for time, ageing is unlikely to be an issue when targeting large (part-per-million level) [CH$_4$] enhancements (Rivera Martinez et al., 2022).

During testing, we characterised each Figaro individually. Previous work has shown that despite using the same Figaro type, individual sensors behave differently (Rivera Martinez et al., 2021, Rivera Martinez et al., 2022) due to variability in sensor surface characteristics (Bastviken et al., 2020, Riddick et al., 2020, Sieczko et al., 2020). Our results plainly show that $R_2$ (see



Table 2) and CH$_4$ response (see Table 4) vary for each sensor. However, some sensors were more similar (for example
LSCE001 and LSCE003) than others (LSCE009), possibly due to batch production with similar surface characteristics. Cho
et al. (2022) applied a single calibration model to 19 different TGS-2611-E00 sensors, but each sensor was assigned a unique
fixed reference resistance. While this was crudely laboratory-tested, with an average 8 ppm [CH$_4$] deviation (sampling up to
190 ppm), it is not clear if this approach was valid in the field (Cho et al., 2022). Elsewhere, Collier-Oxandale et al. (2018)

tested a universal TGS 2600 calibration model, which while promising, could not compete with a sensor specific model,
supporting our approach.

Although our $R_2$ model requires improvement, [CH$_4$] response was characterised very well up to 1 000 ppm in controlled
conditions, with a resistance ratio RMSE of no more than $\pm 1\%$ $\Omega\,\Omega^{-1}$ for the five tested sensors and a R$^2$ of at least 0.997. Our
Eq. (4) [CH$_4$] model is similar to the simple manufacturer-proposed power law (Eugster and Kling, 2012). However, as we

use resistance ratio instead of raw resistance, we include a unity term. This satisfies the requirement that resistance ratio is
equal to one in standard gas (*i.e.* when [CH$_4$] is 2 ppm). Furthermore, Eq. (4) allows other sensitive gases to be multiplicatively
included.

Our Fig. 11 resistance decay curve is similar to the TGS 2611-C00 relationship overserved by Glöckler et al. (2020) up to
9 000 ppm [CH$_4$], although they did not derive a model fit. Honeycutt et al. (2019) proposed a Langmuirian fit in dry

conditions, up to 1 000 ppm [CH$_4$], for various Figaro types. Elsewhere, Rivera Martinez et al. (2021) found a clear resistance-
[CH$_4$] decline up to 9 ppm, but Figaro TGS 2611-E00 resistance changes were less pronounced than for the TGS 2600 and
TGS 2611-C00. van den Bossche et al. (2017) derived a linear TGS 2611-E00 [CH$_4$] calibration, by sampling six [CH$_4$] levels
up to 9 ppm in fixed environmental conditions. Although TGS 2611-E00 resistance appears linear over a small [CH$_4$] range,
non-linearity increases at higher [CH$_4$] (Honeycutt et al., 2019, Bastviken et al., 2020). Cho et al. (2022) derived a resistance

power law up to [CH$_4$] 10 000 ppm, at various temperature settings. Jørgensen et al. (2020) also observed a resistance ratio
power fit up to 100 ppm [CH$_4$]. A similar fit was observed at three different relative humidity settings; however, this model
did not include a unity term as in Eq. (5), meaning that resistance tends to infinity at standard [CH$_4$], rather than a reference
resistance (Jørgensen et al., 2020). A simple power law also limits the model to a single gas.

As Jørgensen et al. (2020) and Cho et al. (2022) targeted emissions where CH$_4$ is the primary reducing gas, their calibration

models only included CH$_4$. We followed a similar approach for our landfill emission source, by simplifying Eq. (4) to Eq. (5).
Alternatively, the TGS 2611-C00 or even the TGS 2600 may be used where only small CO enhancements are expected, as the
lack of a CO filter amplifies CH$_4$ sensitivity (Eugster et al., 2020). In addition, Rivera Martinez et al. (2022) showed that the
TGS 2611-C00 may be less noisy, making it easier to model [CH$_4$] enhancements above the background than the TGS 2611-
E00. This improved TGS 2611-C00 sensitivity may augment an environmental $R_2$ fit. In any case, our Eq. (4) model allows

other gases to be included in future work if necessary. This may allow the TGS 2611-E00 to be deployed in industrial locations
with high CO emissions. However, before considering such an approach, improvements in $R_2$ characterisation are first required.





The small resistance decrease (between 1.4% and 2.0% for the five tested sensors; see Table 4) in response to a 1 ppm [$CH_4$] enhancement above the background, emphasises the importance of accurately modelling $R_2$.

Reference gas testing (Sect. 3.2) revealed that synthetic air and ambient air (from our laboratory), containing the same 2 ppm [$CH_4$], resulted in a different Figaro resistance response. A similar effect may have also contributed towards disparity between landfill Figaro measurements and $R_2$, due to a different air composition in the environmental chamber, compared to the field. A precise gas analysis may identify Figaro-sensitive species in different gas sources, including ambient air at the landfill site (Duan et al., 2020), using techniques such as gas chromatography, Fourier-transform IR spectroscopy or proton-transfer-reaction mass spectrometry, which is particularly suited to detect volatile organic compounds. However, in reality, this would be arduous as it is not clear which interfering gases to look for, especially at a landfill site (Duan et al., 2020). $CH_4$ is the most abundant reducing gas in natural ambient air followed by CO, which were both measured by the Picarro G2401 HPR throughout testing in the environmental chamber and during the laboratory sensitivity test. Although other alkanes (for example, ethane, propane and butane) are reducing gases, with similar chemical properties to methane, they are present in very low quantities in ambient air. Furthermore, manufacturer testing with iso-butane up to 10 000 ppm revealed negligible Figaro resistance response (Figaro Engineering Inc., 2011), though straight-chain alkanes may behave differently. Similarly, alcohols may interfere with SMO sensors, but manufacturer testing up to 10 000 ppm of ethanol also showed negligible Figaro response (Figaro Engineering Inc., 2011). Hydrogen is the only other reducing gas, known to affect the Figaro TGS 2611-E00 (Figaro Engineering Inc., 2011). Maybe different alcohols and alkanes (or some other volatile organic compounds, not discussed here) could play a role, but targeting a specific reducing species, with no obvious candidate, remains a challenge. Unfortunately, it is difficult to look to other SMO prototype sensors to help to identify Figaro-sensitive interfering compounds, as each SMO sensor is unique in its composition and behaviour. Therefore, we recommend a robust analysis of Figaro TGS 2611-E00 gas sensitivities in future work, to help to identify potential interfering gas species in ambient air. In this work, for simplicity, we used ambient air as a standard gas, rather than clean synthetic gas or zero-air when characterising $R_2$. However, this assumes that the air composition during testing was similar to ambient air in the field. A thorough gas analysis may help to confirm this assumption. Alternatively, deploying a field logger containing a suite of low-cost SMO sensors with sensitivities to different gases (including and excluding methane) may help to shed some light on the nature of interfering reducing compounds (Casey et al., 2019, Collier-Oxandale et al., 2019). This test may offer valuable insight into various Figaro sensitivities over a prolonged sampling period.

Another potential cause of $R_2$ disparity between the $R_2$ model and landfill Figaro sampling was the wind dynamics around the System A enclosure, as discussed above. This may be resolved by placing the Figaro sensor in a closed cell more akin to System B. This permits a controlled sensor airflow, resulting in consistent sensor cooling effects. It also buffers temperature changes and allows temperature measurements to be more repeatable in the laboratory compared to the field. This approach would also enable precise gas exposure during environmental $R_2$ testing, rather than relying on potentially contaminated air in and around an environmental chamber. Furthermore, the Figaro sensor would not move between loggers, eliminating the



chance of different loggers potentially causing spurious jumps in sensor behaviour. However, a closed cell logger requires a pump, which has substantially higher power requirements. This may push a solar panel power source to its limits, especially in the mid-latitude winter.

## 6. Conclusion

Ten Figaro TGS 2611-E00 sensors were deployed at a landfill site in France, of which five sensors were tested to characterise environmental and methane mole fraction response. With an ultimate objective of deriving methane mole fraction from sensor resistance, we took the approach of first incorporating environmental effects into a standard reference resistance, thus enabling separate characterisation of gas response. We found that the choice of standard reference gas has a significant effect on Figaro resistance, despite containing the same 2 ppm methane mole fraction: Figaro resistance was much lower in natural ambient air, compared to both synthetic air and a high concentration methane source diluted with zero-air (targeting 2 ppm). We
therefore used ambient laboratory air as our gas standard, which naturally contains 2 ppm methane mole fraction. Temperature and water mole fraction effects were characterised in the field logging enclosure, using a large environmental chamber. Water mole fraction had the largest influence on resistance, but temperature also played a role. In spite of the quality of our environmental chamber model fit, a reference (2 ppm) resistance, as observed in the chamber, could not be replicated in field conditions for a variety of potential reasons. There may have been field differences in airflow around the logger, the air
composition may have been different during testing or there may have been spontaneous sensor variability during transfer between various loggers and in different environments. Nevertheless, our methane gas enhancement characterisation model provided an excellent fit in controlled conditions. We also showed that the effect of carbon monoxide is minimal. Future TGS 2611-E00 work should be conducted with great care, to ensure that environmental effects are well-characterised and that a good choice of standard gas is used, to mirror field sampling conditions. With improvements in a reference (standard gas)
resistance characterisation, it is evident that the Figaro TGS 2611-E00 sensor has great potential in detecting methane mole fraction with a parts-per-million level precision. A closed sampling cell with a pump may help to achieve this goal, although power requirements will have to be taken into consideration.

## Funding

This work was supported by the Chaire Industrielle TRACE, which is co-funded by the *Agence Nationale de la Recherche*
(ANR) French National Research Agency (grant number: ANR-17-CHIN-0004-01), SUEZ, TotalEnergies - OneTech and Thales Alenia Space. This work also received contributions in kind from the Integrated Carbon Observation System (ICOS) National Network France.





**Author contributions**

AS prepared the manuscript in collaboration with GB, PC, OL and EA, who edited the text. GB, PC, OL, RRM and EA provided support and ideas during the testing process. AS designed System B. AS, LL and OL conducted testing of System A and System B and installed System A in the field. EA helped to facilitate site access for field deployment. AS processed testing and field sampling data.

**Competing interests**

The authors declare that they have no conflict of interest.

**Acknowledgements**

We thank Mathis Lozano, Mali Chariot, Timothé Depelchin and Julien Moyé for support in System A field installation and Carole Philippon for support during laboratory testing. We than Pierre Maso, Nicolas Caignard and Sébastien Ancelin at the Plateforme d'Intégration et de Tests at the Observatoire de Versailles Saint-Quentin-en-Yvelines for access and support in using the environmental chamber. We thank the operating staff at the SUEZ Amailloux landfill site for site access, assistance 745 and support. We thank Scientific Aviation, Inc for complimentary System A data access and support.

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

## 925    Appendix A: Influence of supply voltage

The influence of power supply voltage on both resistance and CH$_4$ sensitivity was characterised by testing a Figaro sensor (LSCE009) in System B. $V_s$ was adjusted from the high-precision power supply unit (T3PS23203P, Teledyne LeCroy Inc.) in four tests: test 1 was at 5.00 V, test 2 was at 5.10 V, test 3 was at 5.00 V and test 4 was at 5.10 V. During each test, gas from the zero-air generator was first sampled for at least an hour. Then an ambient target gas cylinder filled with outside air (1.6 ppm
[CO], 2.2 ppm [CH$_4$] and 434 ppm [CO$_2$]) was sampled in four 15-minute intervals. Each ambient target gas interval was followed by 15 minutes of sampling zero-air generator gas. A fixed 8° C dew point was used throughout testing. Gas at this dew point was sampled from at least 24 hours in advance of test 1.

Figaro resistance results for the four tests are presented in Fig. A1. For each test, a 2-minute average was taken at the end each 15-minute ambient target gas sampling interval, except the first (see Fig. A1). A 0 ppm reference resistance baseline was then
derived by fitting a second order polynomial to the final 2 minutes of the each 15-minute zero-air sampling period. [H$_2$O] was on average (0.975±0.001)% during these 2-minute zero-air periods for all four tests, according to the Picarro G2401, and System B temperature was on average (27.9±0.1)° C, according to the SHT85 sensor inside the sampling cell.

The ratio between each 2-minute average ambient target gas resistance and its corresponding modelled zero-air reference resistance was acquired. Each of the four tests yielded three resistance ratios (see Table A1). In addition, for each test, all zero-
air and ambient target gas 2-minute average resistance measurements were combined and averaged in Table A1. These results show that Figaro resistance is consistently lower at higher $V_s$, for example, zero-air resistance at 5.00 V is 35 kΩ whereas at 5.10 V it drops to 31 kΩ. This test also shows that Figaro sensitivity is consistently lower at a higher voltage, owing to a lower resistance ratio. At 5.00 V, the resistance decreases by 22% when transitioning from zero-air to ambient target gas, whereas at 5.10 V, there is a smaller 19% resistance decrease.





| test | supply voltage | average baseline (zero-air) resistance (kΩ) | average target gas resistance (kΩ) | resistance ratios ($\Omega\ \Omega^{-1}$) |
|---|---|---|---|---|
| test 1 | 5.00 V | 35.3±0.3 | 27.7±0.4 | 0.7837±0.0003; 0.7832±0.0003; 0.7824±0.0003 |
| test 2 | 5.10 V | 31.4±0.5 | 25.5±0.6 | 0.8125±0.0003; 0.8105±0.0003; 0.8096±0.0003 |
| test 3 | 5.00 V | 35.2±0.2 | 27.4±0.1 | 0.7796±0.0003; 0.7788±0.0003; 0.7784±0.0003 |
| test 4 | 5.10 V | 31.3±0.6 | 25.3±0.4 | 0.8083±0.0003; 0.8080±0.0003; 0.8080±0.0003 |

**Table A1: Average zero-air and ambient target gas resistances during 2-minute averaging periods for four tests at two different supply voltage settings. The resistance ratio for each 2-minute ambient target gas average is given, compared to a zero-air baseline reference resistance.**

This resistance and sensitivity decrease at 5.10 V emphasises the importance of maintaining a fixed and reliable 5 V $V_s$, to maintain consistency between sensor testing and field application. This effect is possibly due to a higher heater temperature at

higher $V_s$, resulting in lower resistance, as proposed in Eq. (3). Similarly, van den Bossche et al. (2017) found that a 10 mV change in heater voltage resulted in a roughly 1 ppm error in their [$CH_4$] estimate, at constant ambient temperature. However, this does not explain reduced Figaro sensitivity. This sensitivity effect may be caused by a change in the density of electrons within the SMO conduction band under an elevated potential difference.

**Appendix B: Water response delay**

Figaro sensors exhibit a delayed response to [$H_2O$] changes. Fig. B1 shows an example of [$H_2O$] decrease, while a Figaro sensor (LSCE010) continuously sampled gas from the zero-air generator in System B. The dew-point setting was abruptly reduced from 20° C to 8° C, resulting in a 1% [$H_2O$] drop. Sensor resistance appeared to overshoot in response to this [$H_2O$] change and slowly decayed back to a stable level, over many hours. [$H_2O$] was (1.116±0.002)% between 07:30 and 14:30 (UTC), according to the Picarro G2401, while System B temperature was (30.2±0.2)° C, according to the SHT85 inside the

cell, with a small 0.07° C hour$^{-1}$ increase (when applying a linear fit). This negligible temperature change suggests that the observed resistance decay is predominantly an artefact of the water transition. The cause of this effect is not fully understood. It may be related to water desorption dynamics on the surfaces between grain boundaries. Water desorption may not be homogenous throughout the sensor, causing a prolonged delay in reaching a resistance equilibrium. Whereas Rivera Martinez et al. (2021) allowed 35 minutes and van den Bossche et al. (2017) allowed 70 minutes for [$H_2O$] stabilisation, our test shows

that many hours of sampling at fixed [$H_2O$] are needed for sufficient water stabilisation. One full day of constant sampling is recommended.





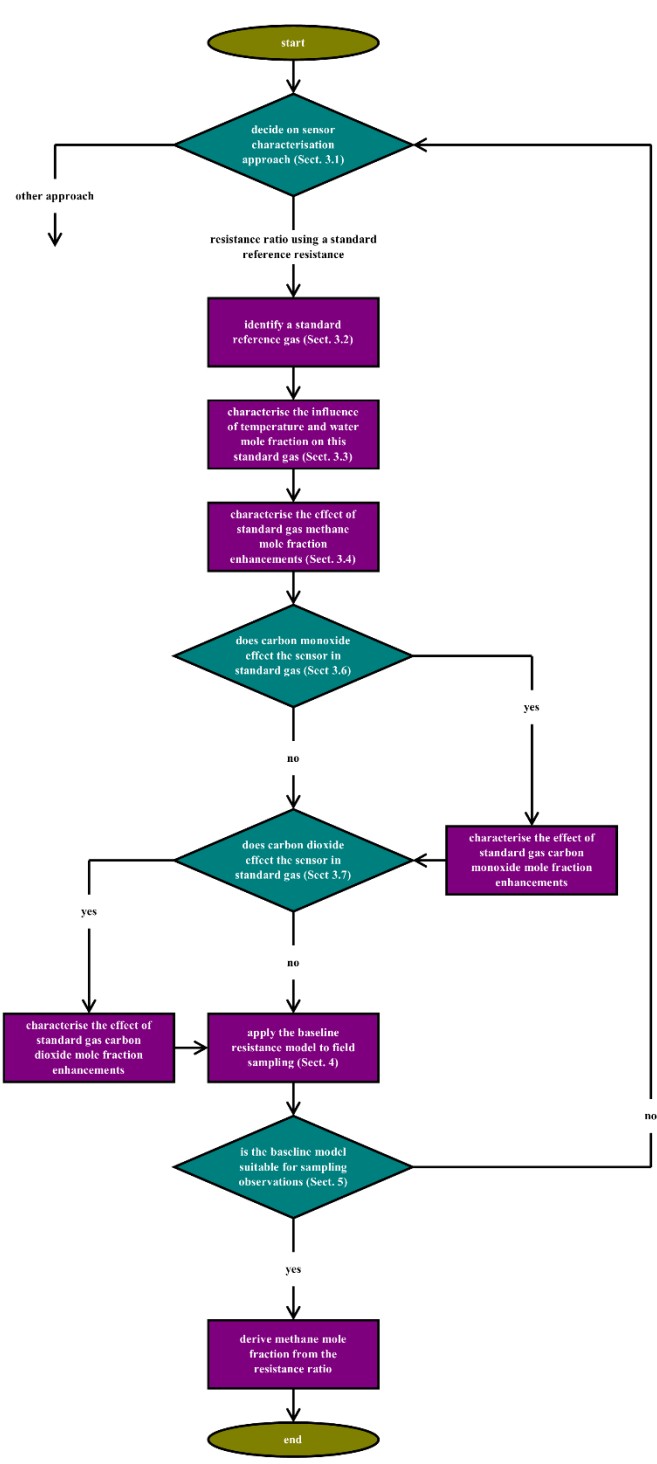

**Figure 1: A flow chart illustrating the various steps that we followed in order to derive methane mole fraction from the Figaro TGS 2611-E00.**



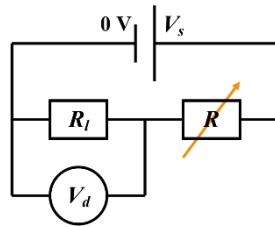

**Figure 2: Circuitry used to measure the resistance of the Figaro sensing element. See text for labels.**

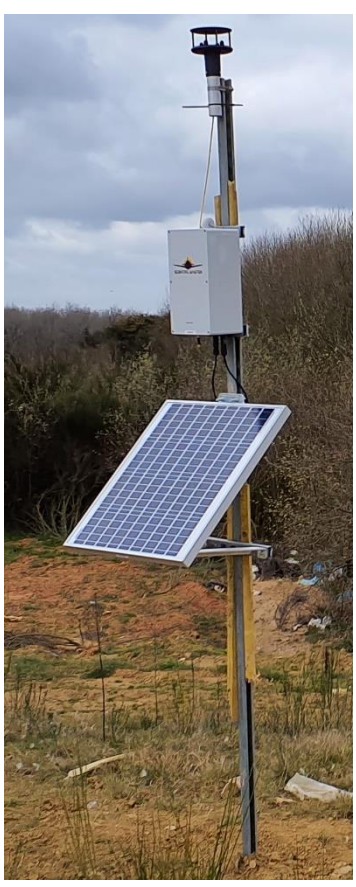

**Figure 3: System A autonomous field logger (LSCE007) installed at the SUEZ Amailloux landfill site in March 2021 (see text for description). This system includes a two-dimensional anemometer.**




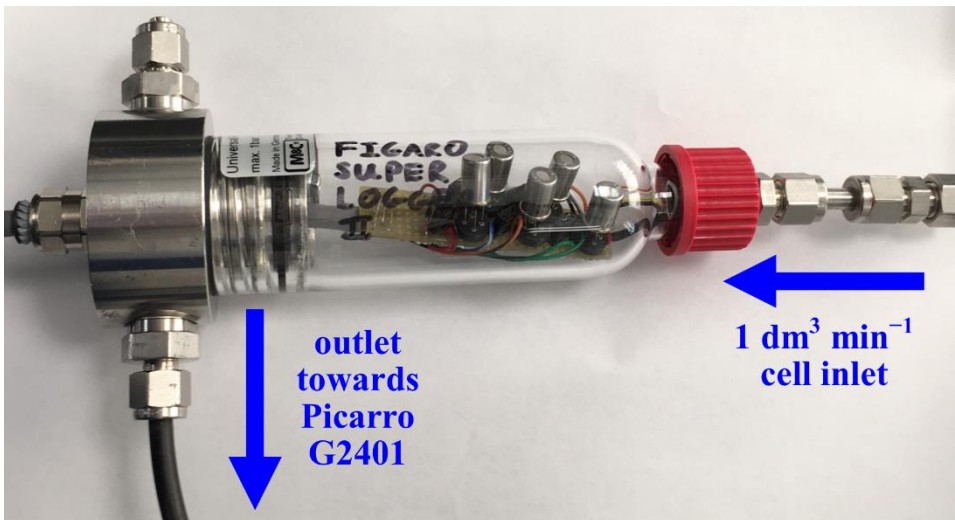

**Figure 4: System B laboratory testing logger. Five Figaro sensors are plugged into the cell circuit board in this photograph.**





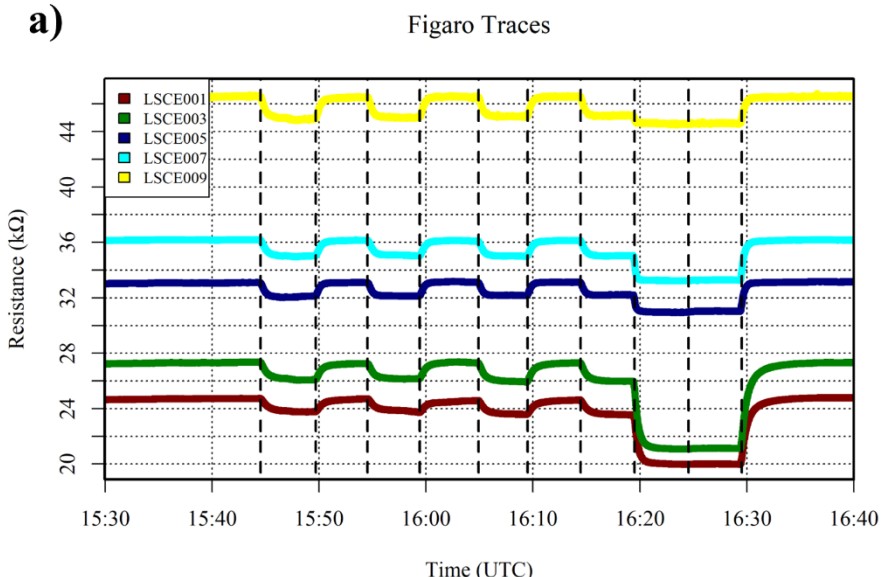

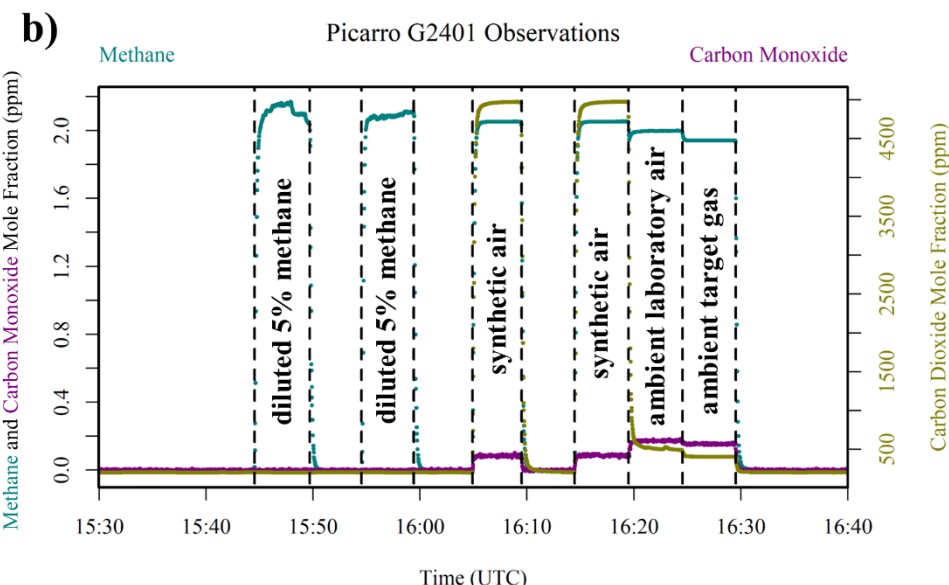

**Figure 5: (a) Measured resistance for five Figaro sensors in System B (coloured dots; see legend) under exposure to various sources of 2 ppm methane mole fraction, compared to gas from a zero-air generator. (b) Corresponding Picarro G2401 mole fraction observations, with annotations indicating the 2 ppm methane source. Areas not annotated correspond to gas from the zero-air generator. Methane (dark cyan) and carbon monoxide (dark magenta) mole fraction measurements are plotted on the left-hand axis. Carbon dioxide (dark yellow) mole fraction measurements are plotted on the right-hand axis.**





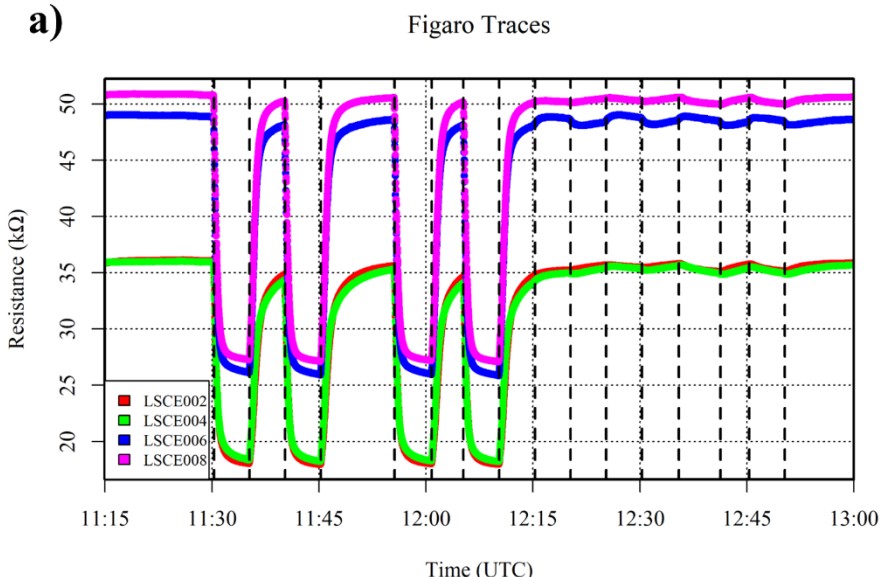

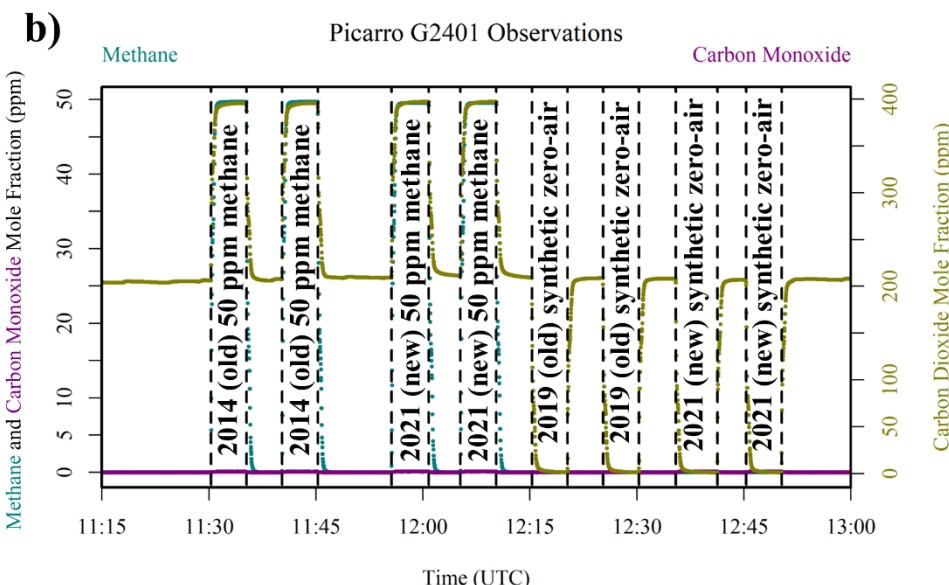

Figure 6: (a) Measured resistance for four Figaro sensors in System B (coloured dots; see legend), under exposure to two sources of 50 ppm methane mole fraction and two sources of synthetic zero-air, compared to gas from a zero-air generator. (b) Corresponding Picarro G2401 mole fraction observations, with annotations indicating the synthetic cylinder type. Areas not annotated correspond to gas from the zero-air generator. Methane (dark cyan) and carbon monoxide (dark magenta) mole fraction measurements are plotted on the left-hand axis. Carbon dioxide (dark yellow) mole fraction measurements are plotted on the right-hand axis.





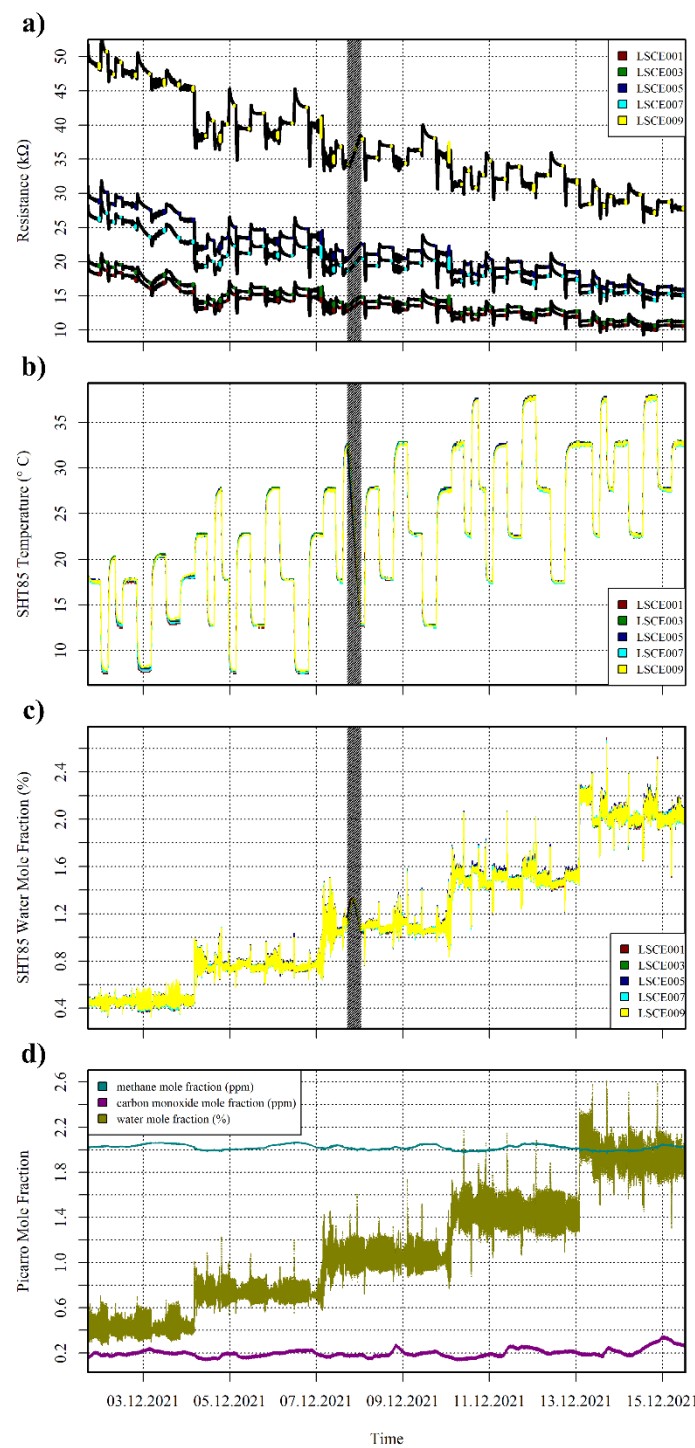

**Figure 7: (a) Resistance for five Figaro sensors, sampling inside the environmental chamber (black dots). Coloured dots indicate data used to derive 30-minute averages for each sampling period (see legend for sensor colours). The shaded area indicates a data**



**transmission gap. (b) Measured SHT85 temperature from each System A box. (c) Derived SHT85 water mole fraction (see text for derivation details) from each System A box. (d) Picarro G2401 measurements from inside the chamber. Methane (dark cyan) and carbon monoxide (dark magenta) mole fraction are plotted in parts-per-million; water (dark yellow) mole fraction is plotted in percent.**



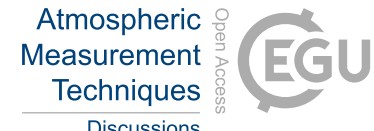

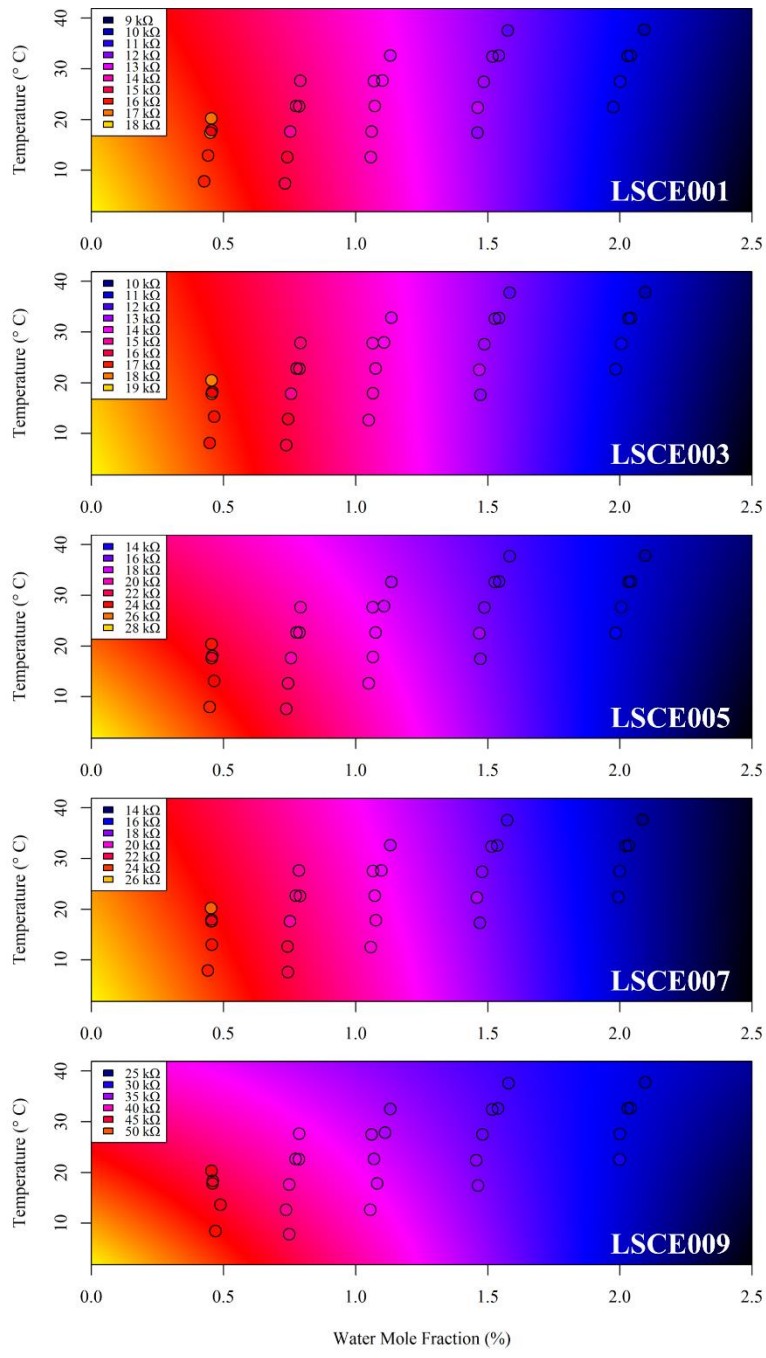

**Figure 8: Modelled reference resistance at 2 ppm methane mole fraction (standard gas) for LSCE001, LSCE003, LSCE005, LSCE007 and LSCE009 (coloured background). Points inside black circles represent 30-minute measured resistance averages. Each plot has a unique colour scale (see legend).**






**a)**

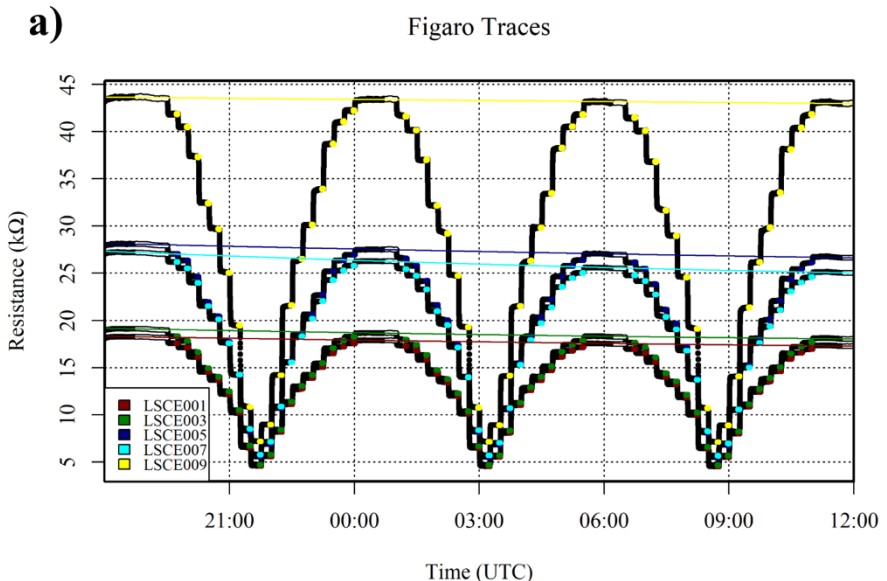

**b)**

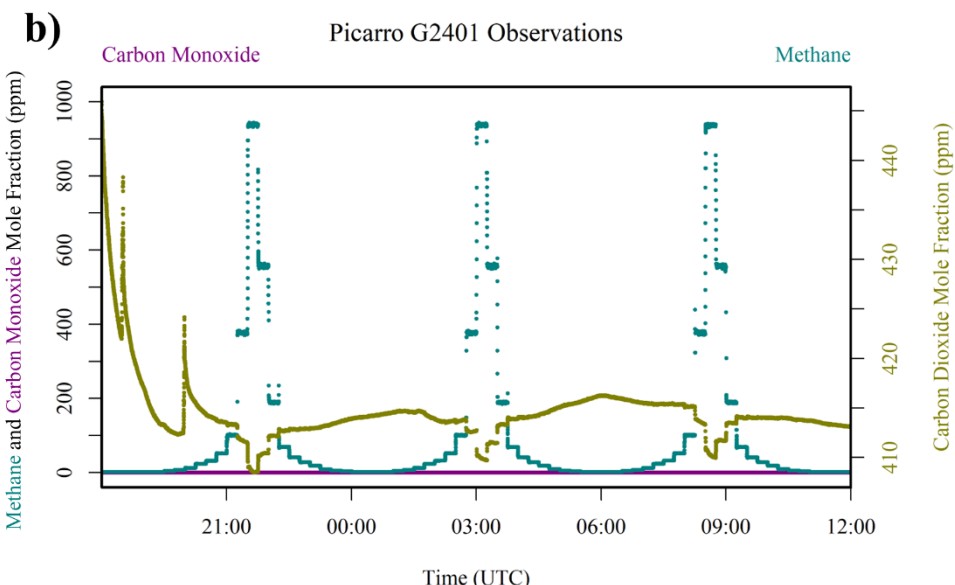

**Figure 9: (a) Measured resistance for five Figaro sensors (black dots), under exposure to various methane mole fraction intervals up to 1 000 ppm. Coloured dots represent 2-minute periods used to derive average resistance values for each methane step (see legend for corresponding sensor colours). White-highlighted dots indicate periods used to derive standard gas reference resistances for each sensor and coloured lines show respective polynomial reference resistance fits. (b) Corresponding mole fraction observations from the Picarro G2401. Methane (dark cyan) and carbon monoxide (dark magenta) mole fraction measurements are plotted on the left-**




hand axis. Carbon dioxide (dark yellow) mole fraction measurements are plotted on the right-hand axis. Carbon dioxide measurements become unreliable at high methane mole fraction due to spectral overlap.

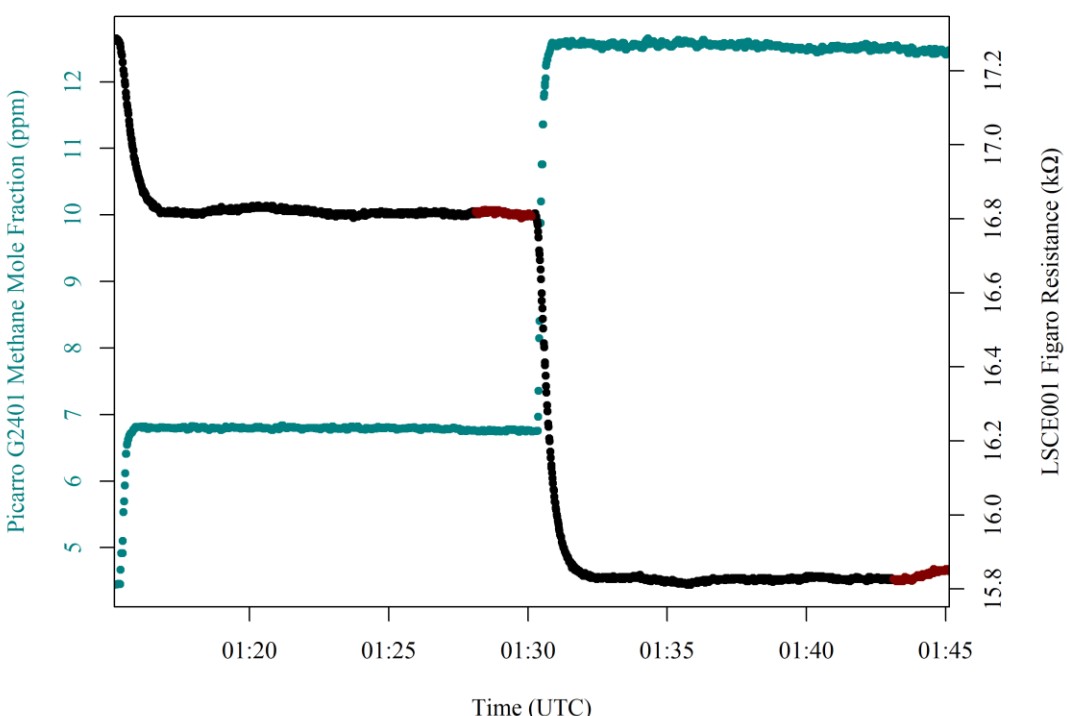

**Figure 10: A methane mole fraction transition from 7 ppm to 12 ppm as recorded by the Picarro G2401 (dark cyan dots on left-hand axis) with corresponding LSCE001 Figaro resistance measurements made in System B (black dots on right-hand axis). 2-minute stable Figaro resistance averages from the end of each sampling period are coloured as dark red dots.**





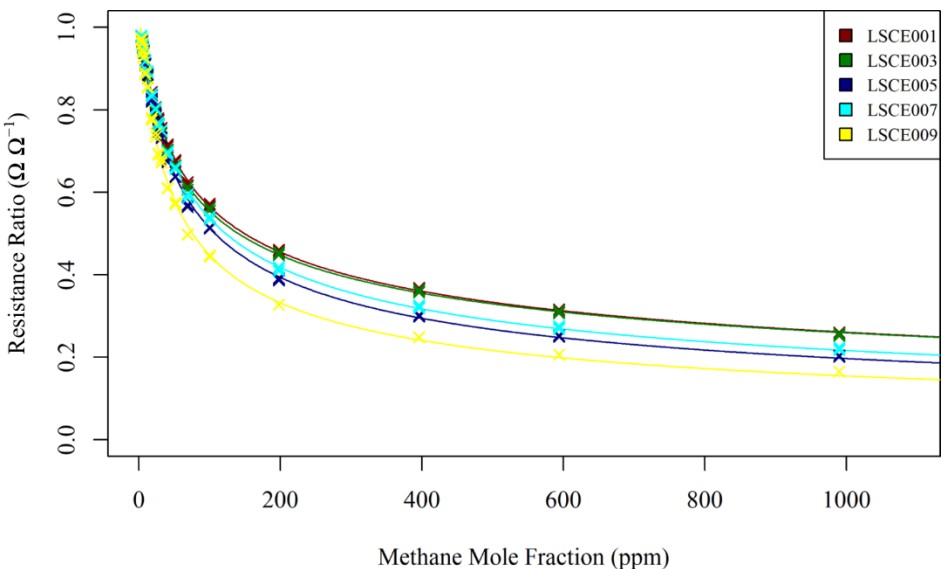

**Figure 11: The ratio between each 2-minute average Figaro resistance (from 15-minute sampling intervals) and its corresponding reference resistance estimate (crosses), plotted against methane mole fraction for five Figaro sensors (see legend for respective colours). A model fit for each sensor (coloured lines) is plotted, according to Eq. (5).**






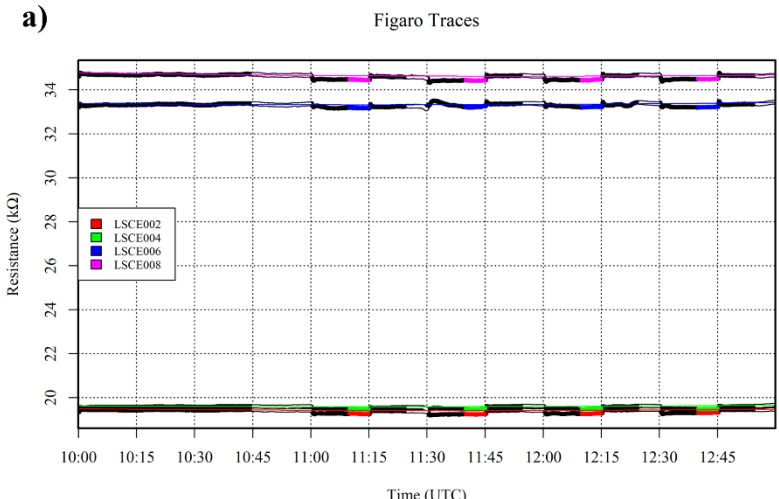

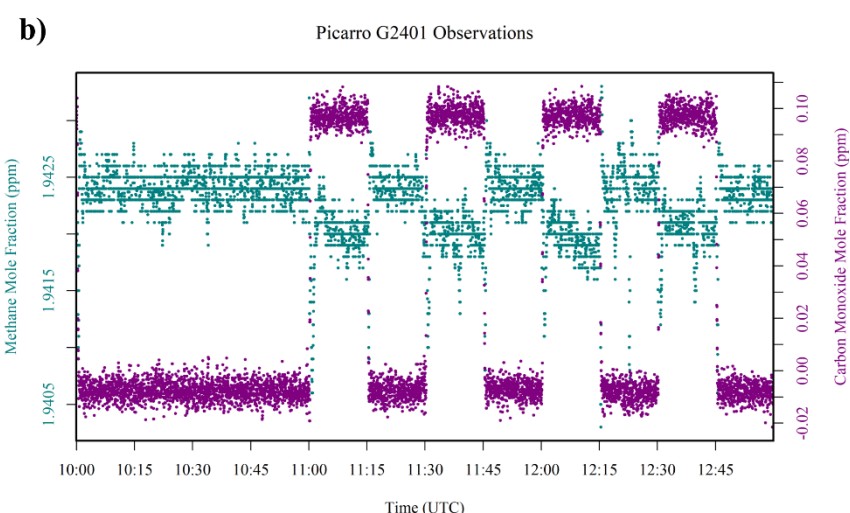

**Figure 12: (a) Measured resistance for five Figaro sensors (black dots), when varying between 0.0 ppm and 0.1 ppm carbon monoxide mole fraction in standard gas. Coloured dots represent 5-minute periods used to derive an average resistance for each 0.1 ppm**

**interval (see legend for corresponding sensor colours). White-highlighted dots indicate periods used to derive 0 ppm reference resistances for each sensor and coloured lines show respective polynomial reference resistance fits. (b) Corresponding Picarro G2401 observations. Methane (dark cyan) mole fraction measurements are plotted on the left-hand axis and carbon monoxide (dark magenta) mole fraction measurements are plotted on the right-hand axis.**





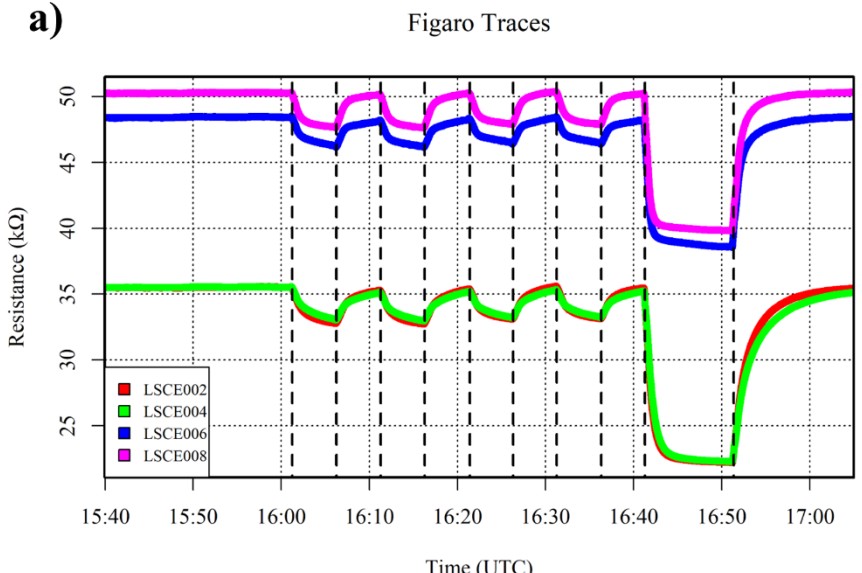

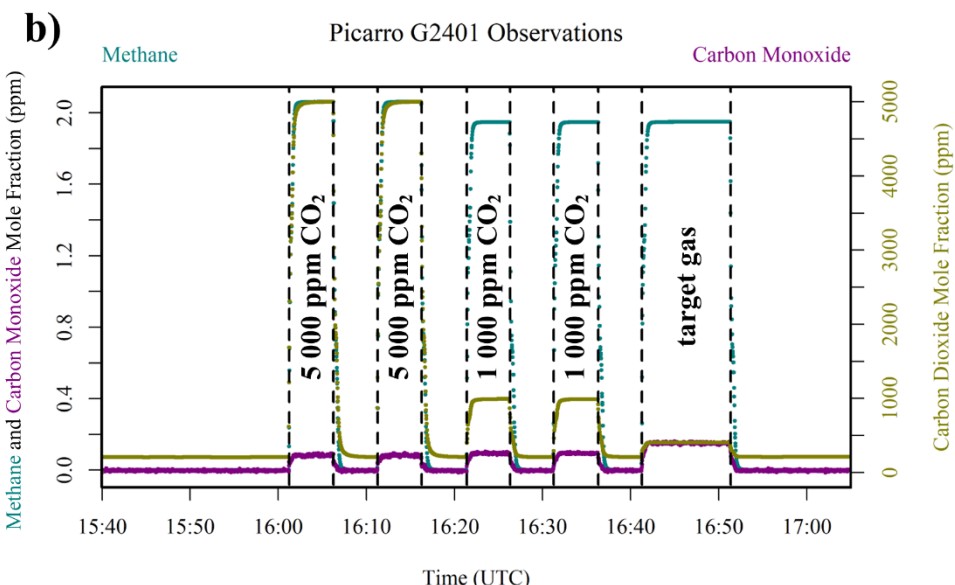

Figure 13: (a) Measured resistance for four Figaro sensors (coloured dots; see legend) in System B, under exposure to 5 000 ppm and 1 000 ppm carbon dioxide mole fraction, compared to gas from a zero-air generator. (b) Corresponding Picarro G2401 observations, with annotations indicating the cylinder type. Areas not annotated correspond to gas from the zero-air generator. Methane (dark cyan) and carbon monoxide (dark magenta) mole fraction measurements are plotted on the left-hand axis. Carbon dioxide (dark yellow) mole fraction measurements are plotted on the right-hand axis.







Figure 14: (a) Measured resistance for five Figaro sensors (black dots), when depleting the oxygen content of gas from a zero-air generator with nitrogen gas. Coloured dots represent periods used to derive 2-minute average resistance value for each interval (see legend for corresponding sensor colours). (b) Figaro resistance averages against corresponding oxygen mole fraction.





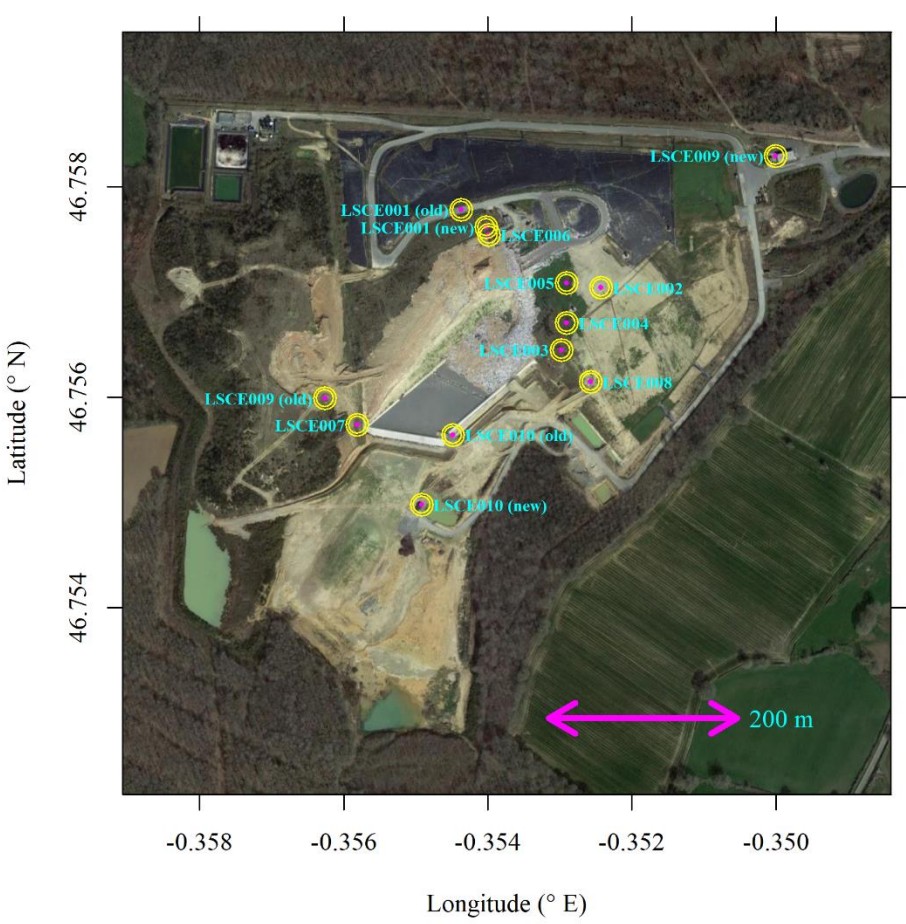

**Figure 15: System A logger locations at the SUEZ Amailloux landfill site. Three sensors were moved from location "old" to location "new" (see text for details). The background image is taken from © Google Maps (imagery (2021): CNES/Airbus, Maxar Technologies).**





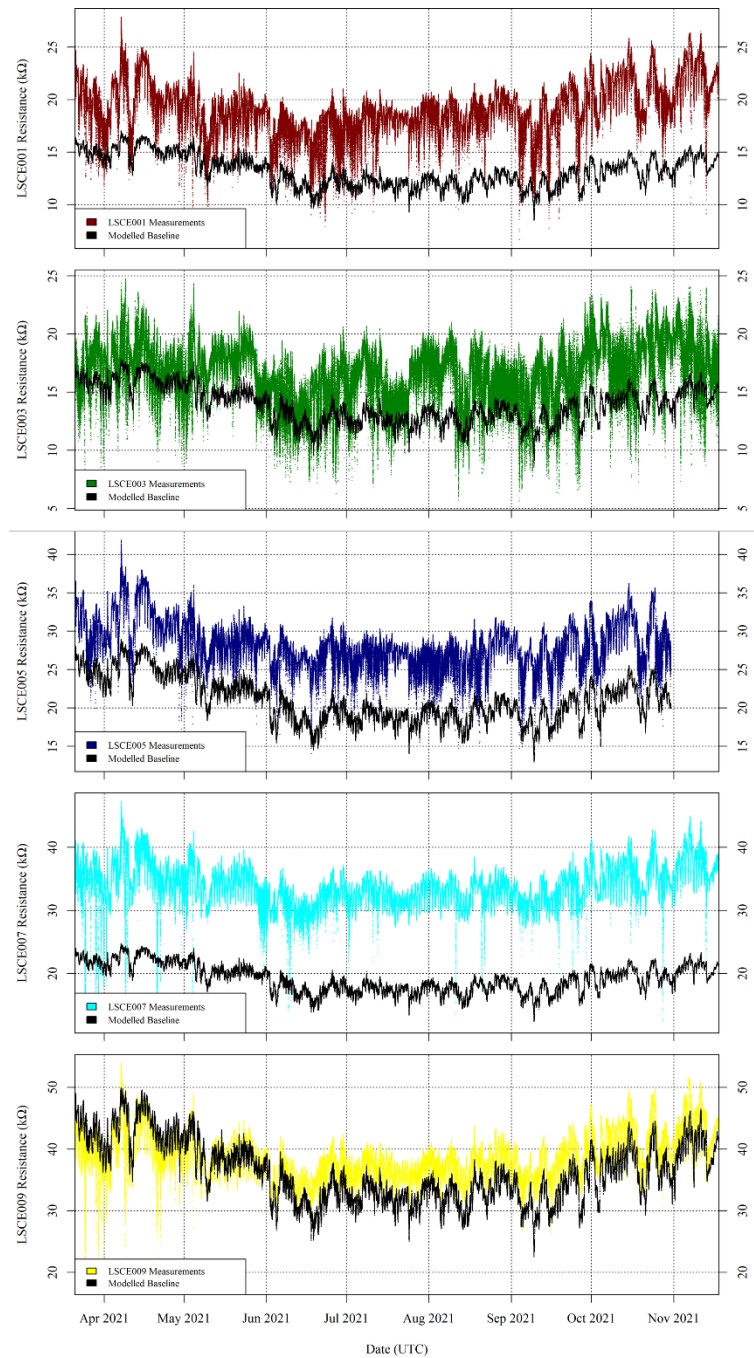

**Figure 16: Measured System A Figaro resistance (coloured dots) and modelled standard 2 ppm [CH₄] reference resistance (black dots) from the SUEZ Amailloux landfill site for LSCE001, LSCE003, LSCE005, LSCE007 and LSCE009 (top to bottom) between 20 March 2021 and 17 November 2021 (period 1).**






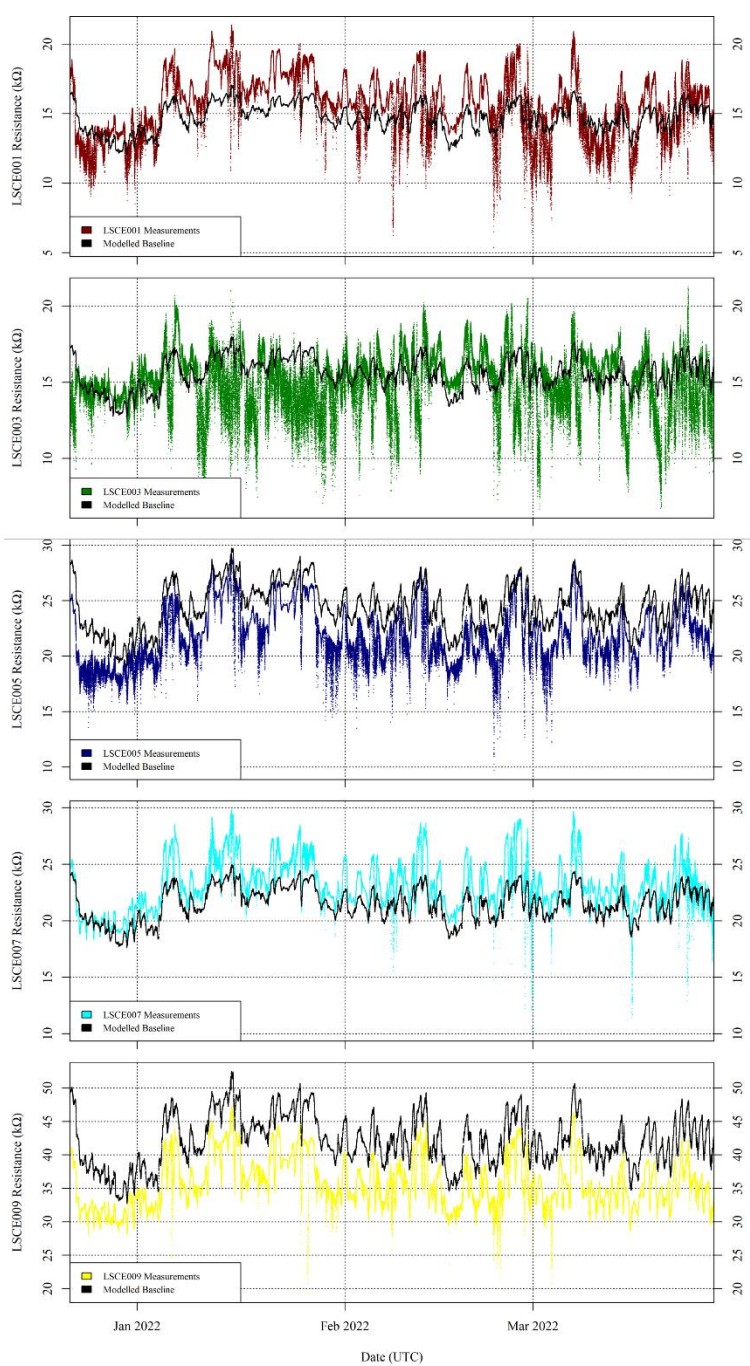

**Figure 17: Measured System A Figaro resistance (coloured dots) and standard 2 ppm [CH$_4$] reference resistance (black dots) from the SUEZ Amailloux landfill site for LSCE001, LSCE003, LSCE005, LSCE007 and LSCE009 (top to bottom) between 22 December 2021 and 27 March 2021 (period 2).**





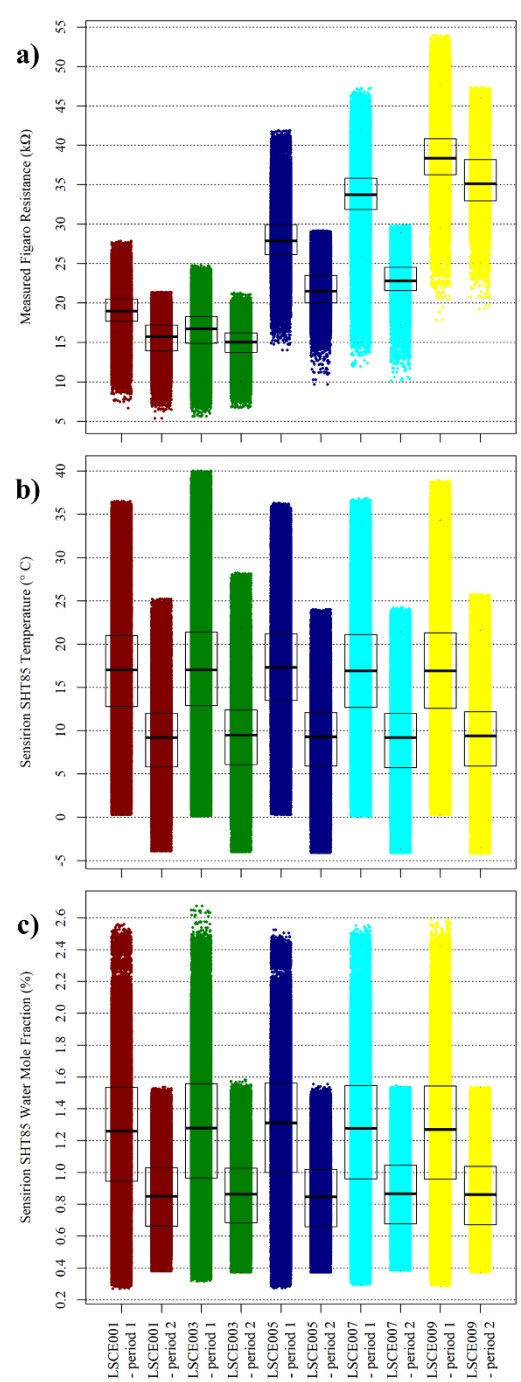


**Figure 18: (a) Measured Figaro resistance, (b) measured SHT85 temperature and (c) derived SHT85 water mole fraction (see text for derivation details), from inside each LSCE001, LSCE003, LSCE005, LSCE007 and LSCE009 System A enclosure at the SUEZ Amailloux landfill site (coloured dots). Data for period 1 and period 2 are plotted separately. The mean and interquartile range for each period are shown in black.**



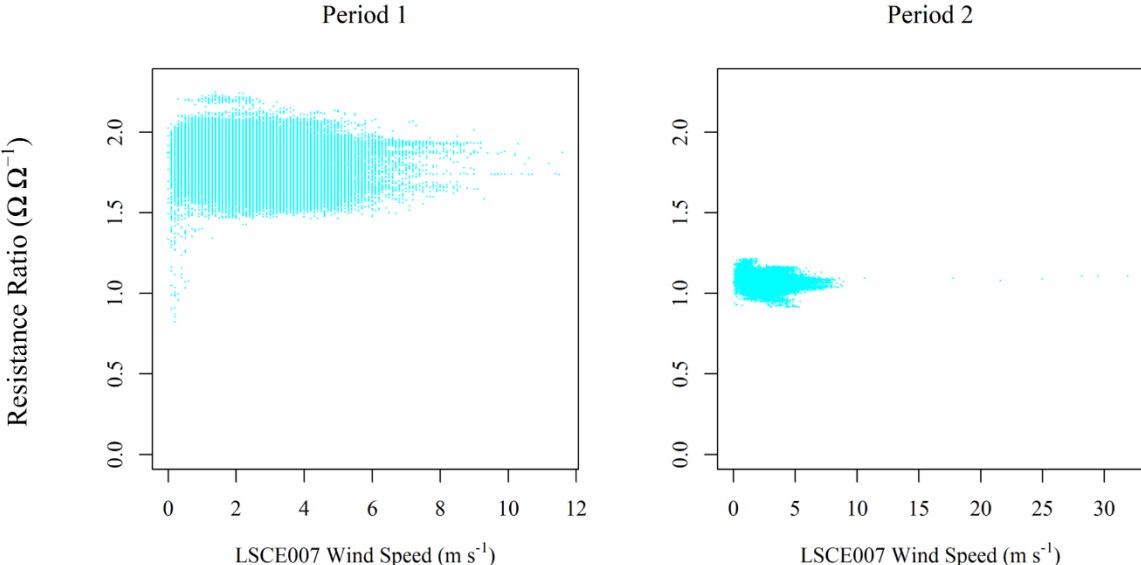


**Figure 19: The ratio between measured Figaro resistance and standard 2 ppm [CH₄] reference resistance (cyan dots) from the SUEZ Amailloux landfill site for LSCE007, plotted against minute-averaged wind speed as measured by the LSCE007 anemometer for wind directions between 180° and 270°. Data from period 1 is plotted on the left and data from period 2 is plotted on the right.**





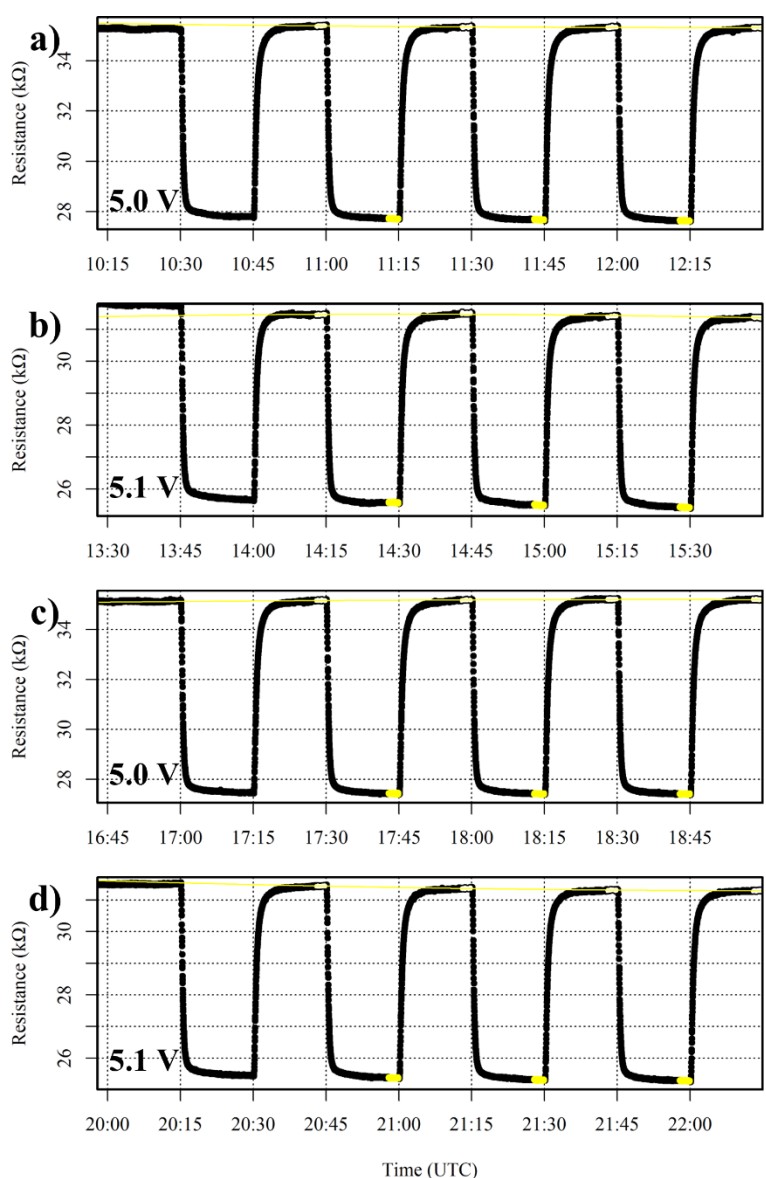

**Figure A1: Measured LSCE009 resistance (black dots), when varying between zero-air and ambient target gas for (a) test 1 at 5.00 V, (b) test 2 at 5.10 V, (c) test 3 at 5.00 V and (d) test 4 at 5.10 V supply voltage. Yellow dots show 2-minute periods used to derive an average resistance value for three ambient target gas sampling periods. White-highlighted dots indicate periods used to derive zero-air baseline resistances and yellow lines show respective polynomial baseline fits.**



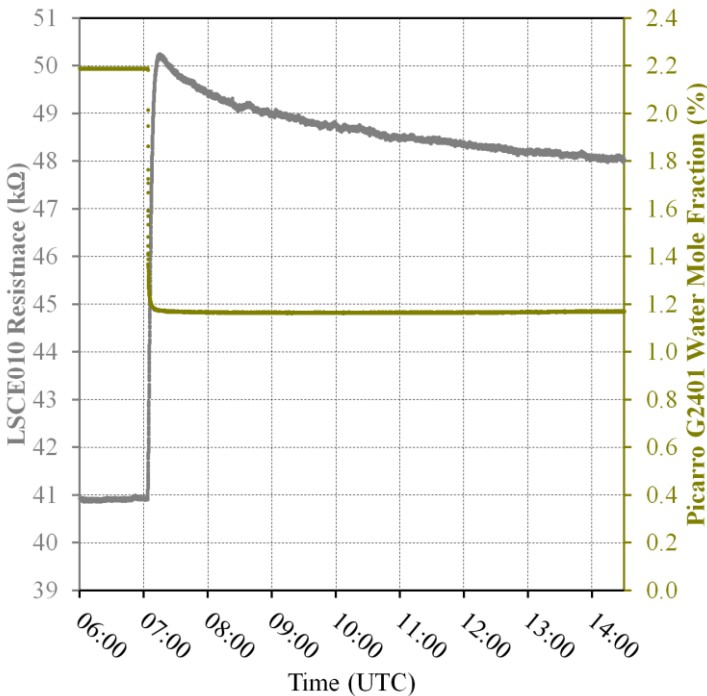

**Figure B1: Figaro LSCE010 measured resistance (grey points; left-hand axis) in response to a water mole fraction (dark yellow points; right-hand axis) drop in System B, as measured by the Picarro G2401, while sampling zero-air generator gas.**