# Peer review of "Characterising Methane Gas and Environmental Response of the Figaro Taguchi Gas Sensor (TGS) 2611-E00"

_Atmospheric Measurement Techniques, 2022_

## Author Comment (AC1)

We thank the reviewer for taking the time to submit a very fair and useful review. This review has greatly helped to improve our manuscript in terms of clarity and our presentation of the key outcomes of this work. We have taken all of the comments on board and have submitted our responses below (red text) to each reviewer comment (black text).

General comments: The paper is a useful addition to the growing body of literature around these low-cost sensors. In particular, the discussion of ambient vs. synthetic air for calibration and the power-law model presented in section 3 are interesting contributions. I found the paper's narrative somewhat difficult to follow, and had difficulty keeping track of the different systems and variables (e.g. System A vs. System B; the various resistance values). Possibly some minor restructuring to center the novel findings and more explicit terminology/notation throughout would help with readability.

We are grateful to the reviewer for their kind comments. We have taken this opportunity to rename certain variables to avoid ambiguity. We have also improved the conclusion section (as suggested by the reviewer) to better emphasise the key findings of this work.

Specific comments by line number:

110-130: While these previous studies report usable results with TGS2600, they used different models and system designs, and it is unclear whether the results are generalizable or tied to specific characteristics of the research sites or experiments. In our previous study (https://amt.copernicus.org/articles/15/5117/2022/) we did not find TGS2600 to respond to low levels of methane in a laboratory setting. This section would be improved by mention of some of the papers' caveats; for example, Eugster and Kling (2012) note in section 3.5 an R^2 of less than 0.20; Collier-Oxandale et al. found different models necessary for their different sites and note that some overfitting was observed, and so on.

We value the reviewer's advice to elaborate on the limitations of the various Figaro testing studies, discussed in the introduction. We have therefore expanded the end of section 1 to highlight specific limitations of previous studies using the TGS 2600. Yet we have remained cautious to limit the level of detail, to avoid this section becoming over-exhaustive.

It is unfortunate that we previously overlooked the interesting outcomes within Furuta *et al.* (2022), due to its recent publication. We thank the reviewer for highlighting its publication and have taken this opportunity to incorporate its valuable findings into our manuscript.

While we recognise that the reviewer did not observe strong TGS 2600 or TGS 2611-C00 methane sensitivity, we suggest that this may be due to dominant effect of variability in [$H_2O$] in their tests. As the reviewer suggests in their own manuscript (Furuta et al., 2022), condensation and evaporation may have occurred from their chamber walls, which we now discuss in section 5.2. In fact, we encountered a similar issue during our own chamber testing work, when we first conducted tests with humidity control turned off. For this reason, we had no choice but to use humidity control, resulting in vast variations in [$H_2O$] (see section 3.3), albeit about a central targeted humidity level.

172 & 184: The 5k resistor choice needs justification. As you note at line 165, the reference resistor should be close in value to the expected sensor resistance. Around background methane levels one would expect a sensor resistance an order of magnitude larger than your

5k choice, as can be seen in your supplemental information (which is consistent with our observations).

The reviewer is right to suggest that a load resistor with a higher resistance would have been more appropriate in this work, based on typical resistance measurements made by the Figaro sensors. However, we chose to use the same load resistance in both System A and System B, to make measurements and systematic errors from both loggers as comparable as possible. Thus, we were effectively bound by the resistance selected by the System A manufacturer.

Therefore, we now clarify that the System A load resistor was hard-wired into the logging box in section 2.2. We now also state that we chose the load resistance in System B to mirror System A, in section 2.3.

The choice of a 5 k$\Omega$ load resistor used by the System A manufacturer can be justified by looking to load resistors used in pervious work. For example, Jørgensen *et al.*, 2020 used a 10 k$\Omega$ resistor. We now make this additional point in the manuscript in section 2.2. We also include a short discussion on this point in section 4.2, where measurements from the Amailloux landfill site are presented. Although a higher load resistance would improve sensitivity, 5 k$\Omega$ is sufficient to detect resistance variations in response to changes in the surrounding environment.

169-179: More detail about the logging system would be useful, particularly the ADC resolution, noise floor, and so on, as these can be expected to determine sensitivity in combination with the reference resistor choice. I would be interested in a brief sensitivity calculation using the ADC resolution/noise floor and the 5k reference resistor to show the ability to detect small changes in the sensor resistance. The logger's power supply stability is also a critical detail for your system's performance.

This is a useful suggestion. We have provided additional details on the ADC resolution in section 2.3. We now specify that although it has a maximum resolution of 18 bit, we set it to sample at 16 bit (for improved sampling frequency), resulting in 0.15 mV resolution. This translates into a resistance precision of 0.6 $\Omega$, assuming 5 k$\Omega$ Figaro resistance.

According to the manufacturer data sheet, the System B power supply has rated ripple and noise effects of less than ±1 mV. In reality, we observed a standard deviation of ±0.1 mV, when tested with the ADC. We now also provide this additional detail in section 2.3.

We thank the reviewer for the suggestion to include these details, which we previously overlooked in terms of importance. We hope that these useful resolution values support our measurement setup.

194: For the application 35mV is possibly not good enough; is this value referring to the accuracy of the setting, ripple, drift, or a combination of all of them? 35mV ripple would swamp any sensor response to small methane changes, for example. You have some discussion of supply voltage sensitivity in your supplemental information, but a short quantitative discussion of what this supply voltage tolerance means in terms of sensor response/detection capability would be illuminating, as it's not immediately obvious to me how big of a concern the supply voltage accuracy is for your experiment.

We agree that an actual 35 mV power supply accuracy (including ripple effects) would be insufficient in this work. In reality, this 35 mV value refers to a read-back setting accuracy, which we now clarify in the manuscript in section 2.3. As we adjusted the supply voltage using an independent measurement of potential difference across the Figaro circuit board, the power supply voltage accuracy has no effect on our testing. To further augment our confidence in the stability of our power supply unit, we measured the circuit board potential difference on many occasions on different days, finding no change in voltage. Nevertheless, the reviewer raises an important point here which we hope we have now resolved in the manuscript in section 2.3.

311-318: The H2O fluctuations within each period are substantial (from the chart, up to 0.5%) and appear to continue over the whole of each sampling period. If the sensors require hours to stabilize at a given humidity level, will they stabilize at all given this large fluctuation?

The reviewer makes a valuable observation regarding the large fluctuations in $[H_2O]$ during our environmental chamber tests. Our work shows that the key issue associated with unstable water, is the associated lengthy Figaro stabilisation time, whereby resistance gradually exponentially decays towards a stable resistance (see Appendix B). We showed that this resistance decay occurs immediately following a step-change in $[H_2O]$.

In the case of the environmental chamber, rather than a single step transition in $[H_2O]$, there is periodic variability about a central point. This phenomenon occurred because the humidity control system constantly works to rectify humidity towards a target setting. As this water variability occurred both above and below the target $[H_2O]$ setting, the water resistance delay effect effectively occurred in both directions simultaneously, thus cancelling itself out.

This can be observed in the top pane of Fig. 7, where after the initial temperature change, Figaro resistance appears to stabilise within 8 hours. Thus, resistance delay effects due to changes in $[H_2O]$ cancel each other out. This argument is now explicitly clarified in the manuscript, where suggested by the reviewer in section 3.3.

316 and throughout: Could you remind the reader what R2 indicates or use a more descriptive subscript? I had difficulty remembering the resistance notation, much of which is similar - R2, Rb, Rl, etc.

To reduce ambiguity, $R_l$ has been renamed to $R_{load}$, $R_b$ has been renamed to $R_{baseline}$ and $R_2$ has been renamed to $R_{2\,ppm}$. We hope that these changes make these different resistance terms easier to follow.

337: As the sensors are much more responsive to humidity than to methane, is 3% uncertainty good enough?

This 3% uncertainty value represents the uncertainty in the $R_{2\,ppm}$ resistance baseline (at 2 ppm $[CH_4]$). $R_{2\,ppm}$ is calculated as a first step, before subsequently calculating $[CH_4]$ enhancements above the 2 ppm $[CH_4]$ level. As $R_{2\,ppm}$ and $[CH_4]$ are calculated separately, and as different steps, it is difficult to compare the two uncertainties and to aggregate them. Therefore, it is not straightforward to make a direct link between uncertainty in $R_{2\,ppm}$ and a theoretical resulting uncertainty in $[CH_4]$. Each uncertainty for each model must be treated independently. We now emphasise the significance of this 3% value more clearly in the manuscript in section 3.3, to avoid confusion.

As we were unable to derive [CH$_4$] estimates during field deployment, it is difficult to know whether this 3% uncertainty in $R_{2\,ppm}$ had a significant influence on the overall [CH$_4$] uncertainty.

355: Fig 10 shows some fluctuation in sensor response in the last two minutes, and it looks like the sensors stabilize at the new methane levels quite quickly. Why is it better to select the last two minutes of each methane level rather than the last 10 minutes, which appear to already be stable?

Our original Figure 10 was used to highlight the general stability of our logging system, rather than the stability of individual Figaro sensors. This figure did not therefore well-illustrate Figaro stabilisation issues, as LSCE001 (shown in the original figure) in one of the more stable sensors that we tested. On the other hand, LSCE009 took longer to stabilise in response to changes in [CH$_4$]. We have therefore updated Figure 10 with an example of a LSCE009 methane transition, which shows a longer stabilisation period.

Based on the delayed stabilisation effect illustrated in this figure, we used a 2-minute averaging period for all five sensors, for consistency. Furthermore, Figure 11 justifies our choice of averaging time, as the [CH$_4$] points fit the model curve very well. This satisfies the ultimate aim of this test to characterise methane sensor response. In theory, taking the final minute instead of the final two minutes, would probably produce the same result with no advantage either way. We have added a sentence to section 3.4, to support this reasoning.

361: Why is Eq. 3 only valid for system A?

We thank the reviewer for raising the ambiguity in this statement. In fact, Eq. 3 is an entirely empirical logger-specific model used to relate measured temperature and derived [H$_2$O] to measured resistance. However, Figaro resistance is also influenced by logger-specific parameters such as airflow and thus associated cooling effects. Furthermore, the gradient between the point of each temperature measurement and each Figaro sensor is also logger-specific, especially for System B, where five different Figaro sensors were tested in the same logging cell. Therefore, Eq. 3 model parameters can only be used for the logger in which they have been derived. We now clarify these points in section 3.4 and section 3.3.

Table 4: The variation in the alpha values for the sensors is surprising to me - our previous work found TGS2611-E00 to be quite consistent, at least within the same production batch. Were your sensors taken from the same batch, or is there some other component in the system that might be causing this variation? You mention this at line 641, but it would be good to also indicate whether your sensors have the same or different batch codes (printed on the side of the component).

Unfortunately, it is difficult to satisfy this point directly at this moment, as the sensors are currently deployed in the field. However, we know that LSCE001, LSCE003 and LSCE005 were purchased by Scientific Aviation in the USA, whereas LSCE007 and LSCE009 were purchased by us at a later date in France. So, we are fairly certain that at least two different batches were tested, but there may have been more. We now add the point that the sensors likely come from at least two batches in section 5.2 of the manuscript. Unfortunately, if is difficult for us to elaborate any more on this without certainty on specific sensor batches.

421: Again, why the last five minutes? It looks like the sensors stabilize more quickly than that, as far as I can see from Fig. 12.

The averaging duration was chosen as a compromise between maximal stability and maximal averaging points. As the sensors stabilised much faster in this carbon monoxide test compared to the methane test (see response above), a longer averaging period of 5 minutes was more appropriate here, than 2 minutes used for the methane test.

We now highlight this point in section 3.5 of the manuscript and explain our rationale for selecting a 5-minute averaging period. In any case, we do not feel that prolonging the averaging time would have a significant impact on the outcomes of this test: choosing a 10-minute averaging time instead of a 5-minute averaging time would probably yield the same qualitative conclusion.

Section 5.1: Your field and lab tests presumably used different power arrangements. Could you add some discussion of the steps taken to ensure consistent electrical operating conditions between the field and lab tests (particularly 5V supply stability)? In the supplemental information you show that different supply voltages cause different sensor responses; is this possibly involved in the differences?

The reviewer makes an interesting point which we previously failed to clarify. The System A logging system converts battery voltage into a stable 5 V Figaro power supply over a wide battery supply voltage range. Therefore, despite the 12 V lithium ion phosphate battery being connected to a charger instead of a solar panel, the Figaro power supply remained unaffected. We now make this point clear in section. 5.1 and also in section. 2.2, where we introduce the System A field logger.

Section 6: This section is difficult to read to me, and doesn't highlight the major contributions of the paper. It would be more clear to me if broken into multiple paragraphs, and with stronger emphasis on the findings you believe to be particularly important.

We appreciate this useful suggestion and have broken section 6 up into smaller paragraphs. We have rephrased the existing points in this section to improve clarity. We have also included additional key points, to emphasise the key findings of our work, as suggested by the reviewer. For example, we now stress the value of the laboratory testing results, where we characterised methane response using an adapted power fit.

---

## Author Comment (AC2)

We thank the reviewer for taking the time to read our paper and for making a number of useful suggestions to improve the manuscript. This review has helped to reduce ambiguity in the manuscript and improve the presentation of our work. We have taken all of the comments on board and have submitted our responses below (red text) to each reviewer comment (black text).

"Characterising Methane Gas and Environmental Response of the Figaro Taguchi Gas Sensor (TGS) 2611-E00" introduces two field and laboratory logging systems (System A, System B), which were used for an extended deployment at a landfill site with Picarro reference and in an environmental chamber setup, respectively. Resistance of the Figaro sensor in relation to methane, temperature, and water vapor were modeled. Both phases resulted in unexplained variability (ambient air vs synthetic air under lab conditions; and ambient air resistance model in field conditions). A thorough discussion of the issues, including literature discussion, is presented. I think the work is sound and have only minor comments:

We appreciate that the reviewer recognises the value of our work. We hope to have improved the manuscript further by following their suggestions.

L88 Is 'trialed' meant here instead of 'trailed'?

We thank the reviewer for spotting this error which we have now rectified, as suggested.

The references to needing to operate in 'wet air' (L111) and elsewhere should be clarified. The Riddick et al. 2020 paper specifically talks about high uncertainty below 40% relative humidity. Additionally [H2O] is defined as the 'water mole fraction' on L19 but that this is usually specified as water vapor mole fraction since solid and liquid phases are also possible.

We agree that greater clarity is required here. Having reviewed the cited literature again, we have improved the details provided on the effect of water vapour on sensor behaviour. Rivera Martinez $et$ $al.$ (2021) showed that resistance was abnormally high at 0% [$H_2O$] compared to 1% [$H_2O$]. Eugster and Kling (2012) showed that TGS resistance was unpredictable below 35% relative humidty. Riddick $et$ $al.$ (2020) remarked that based on the Eugster and Kling (2012) study, calibrations must be performed in wet conditions. Following our changes, we hope that these points are now more accurately incorporated in the manuscript in section 1.

We have also now changed "water mole fraction" to "water vapour mole fraction" to avoid confusion and to make it clear that we are referring to water in the vapour phase.

Fig 4: Is the Raspberry Pi 3B+ computer outside of the cell pictured? This is important for understanding the data logging configuration

This is a good point. We now clarify in the caption for Figure 4 that the photograph only depicts the logging cell and not the entire logging system, with the logging computer and power supply not shown. The figure has been updated with an arrow pointing towards the cable which both supplies power and provides connections to the analogue-to-digital converter. We also now explicitly clarify the external placement of the logging computer in section 3.3.

L200 Is Picarro serial feed sent to the Raspberry Pi? It is minor, but simultaneous logging on a computer does not automatically solve timing issues because the Picarro and ADC board used for the Figaro still have separate clocks.

The Picarro streams data to System B using a serial data connection. This detail has now been added to the manuscript. When the Picarro data reaches the System B logger, it is written directly into the System B data file. Therefore, only the System B timestamp is used. There is no use for the Picarro timestamp, which is not recorded by System B, so it does not matter if the Picarro has a separate clock. We apologise for any confusion here and have made efforts to make this point clearer in section 2.3.

As the Picarro data is simply written into the datafile alongside the Figaro data, it does not matter whether the time is accurate or not. So long as a single time stamp is being used, the laboratory experiment is not affected. We acknowledge that this does not solve timing accuracy issues (*i.e.* if System B time in not precisely equal to UTC). However, it eliminates any issues with a time offset (*i.e.* if the Picarro time is different to the System B time), which may become an issue if attempting to combine two separate data files.

L233 suggest moving the statement "All synthetic air cylinders contain a natural balance of nitrogen, oxygen and argon." to the paragraph starting L254 once it is mentioned that multiple synthetic air cylinders were used

This is a good idea. We have followed the reviewer's suggestion.

L239 Does stabilization of [H2O] in the 'large environmental chamber' also play a role, or the settling can simply be attributed to the Figaro?

As this test was conducted with System B in the laboratory and not in the environmental chamber, we can be certain that the observed effect was due to the Figaro itself. We apologise for any confusion here and added a sentence in section 3.2 describing how System B testing was performed in an air-conditioned laboratory. We have also added a sentence in section 2.3 to clarify that this test took place in System B, where [$H_2O$] was held constant, thanks to the dew-point generator. Only System A was tested in the environmental chamber.

L345 How was the dilution from 5% [CH4] in argon all the way down to 2 ppm achieved? The accuracy of the dilution seems somewhat important, since the agreement with the 2 ppm synthetic air and disagreement with ambient laboratory air is a key area of discussion in the manuscript

In this methane characterisation test, ambient air was used as a standard reference gas, which naturally contains about 2 ppm [$CH_4$]. Therefore, 2 ppm [$CH_4$] was quite simply achieved by sampling pure ambient air, with no dilution. In order to increase [$CH_4$] up to 1 000 ppm, we used mass-flow controllers to add small quantities of gas from a 5% [$CH_4$] cylinder. We now direct the reader to section 2.3 in section 3.4, where the mass-flow controllers are discussed. The Picarro G2401 reference instrument was used to deduce the resulting [$CH_4$] from the gas blend.

In section 3.2, where different 2 ppm [$CH_4$] sources were compared, a 2 ppm [$CH_4$] sample was achieved by diluting 5% [$CH_4$] gas with zero-air generator gas, which contains 0 ppm [$CH_4$]. This was also achieved using mass-flow controllers. As all Figaro laboratory testing

was conducted alongside a Picarro G2401 reference instrument, we can be sure that the mass-flow controllers successfully produced a 2 ppm gas blend during this test. Thus, we were always aware of the [CH$_4$] level in the gas stream, regardless of uncertainty in the mass-flow controller flow rate.

L522 'model yielded excellent R2 agreement during chamber testing (see Fig. 8)' This is not shown in Figure 8

The reviewer is right to highlighted that Figure 8 is a visual representation of the background resistance model and does not allow the reader to evaluate model agreement. We instead direct the reader to Table 2 here, where R$^2$ and RMSE values are given.

L568 I think the statement 'chamber testing may not be suited for SMO sensors in general.' is stronger than what is said in Eugster et al. 2020. Moreso that lab calibration can, and ideally should, be incorporated, but field calibration is simpler to do accurately

This is a good point. We have rewritten this sentence more factually, without drawing general conclusions from the work of Eugster *et al.* (2020). We simply state that Eugster *et al.* (2020) yielded unsatisfactory results from chamber testing, as stated in their section 3.5.

The figure font sizes / arrangement could use some work, as Figures 1, 7, and others are relatively hard to read even while spanning a full page in this version

We have updated most of the figures in the manuscript, with improvements including higher resolution and fewer white spaces. Regarding Figure 1, we have changed the background colour to white, making the text easier to read, which we have also emboldened. We have made a number of improvements to Figure 7 including increasing the size of the axis labels and titles. We have also plotted all environmental chamber resistance measurements as coloured dots, instead of black dots, to make them easier to distinguish. The periods used to derive 30-minute averages are now shown as black bars at the top of the plot. In the previous version of this plot, individual SHT85 temperature and water mole fraction measurements from each System A logger were plotted as overlapping dots. We have now changed this by presenting average temperature and water mole fraction values from all five System A boxes, as they are almost identical. The average standard deviation in temperature was 0.14° C and the average standard deviation in [H$_2$O] was 0.01%, between the different System A boxes.

On Fig 18, the number of data points is high so using a stripplot for the individual data does not add much information versus just showing a standard boxplot. A swarmplot may be preferable for showing the individual points

The reviewer makes a good suggestion. We attempted to produce a swarm plot here, but there were too many densely packed measurements in the centre of each data range, making it impossible to present this figure nicely. However, we fully agree with the reviewer's suggestion that a box plot is otherwise more apt here. We have therefore updated this figure as a simple box plot.

---

## Author Response (AR2)

Dear Editor,

We appreciate the suggestion to remove some figures from the main manuscript to a new supplement. We agree that the essence of the paper would not be affected with the loss of a number of figures and that the readability of the paper would improve. All changes to the manuscript are highlighted as track changes (see attachment).

We have decided to move Figure 1 to the supplement, which is a flow diagram of the steps to derive methane mole fraction from a Figaro sensor, in an ideal case. Although this figure is useful to envisage the various sections presented in our work, an adequate description is given in words at the end of section 1. This large figure is therefore not immediately necessary to the reader to understand our testing procedure.

We agree with the Editor that Figure 2 should also be moved from the main manuscript, which shows a circuit diagram of the Figaro sensor, its corresponding load resistor and the power source. It is worth including it in the supplement as a figure of reference, so that the reader can understand how the Figaro sensor is integrated into a circuit. But for most readers with expertise in atmospheric science, this sort of technical detail on electronic circuitry in not relevant.

We recognise the Editor's suggestion to remove Figure 3 and Figure 4 from the main manuscript and agree that the loss of these figures would not impact the conclusions of our work overall. However, we believe that these figures provide significant added value by allowing the reader to picture the logging set-up. For example, Figure 3 provides a visual representation of sensor installation in the field which is difficult to capture in words. Meanwhile, Figure 4 is useful in order to visualise the laboratory testing set-up. This view is supported by Reviewer 2 who requested greater clarification of Figure 4, which we then included.

The next figure that we have decided to remove is Figure 10, which shows an example of a transition in methane mole fraction as recorded by both the Picarro G2401 reference instrument and a single Figaro sensor. This figure is used to emphasise the stability of the sensors. It also shows the time delay for Figaro stabilisation in response to methane mole fraction changes. As this figure is effectively an extract of Figure 9 (showing the full test for all five tested sensors), it does not need to be in the main manuscript and can be available in the supplement, to support the conclusions made in the main text.

Regarding the Editor's suggestion to remove Figure 17, we recognise their point that this figure appears very similar to Figure 16. Both figures show that the baseline resistance model cannot be used to model resistance in the field. However, it is only through comparison of the two figures that the significant period 2 decrease in measured resistance (compared to modelled resistance) can be appreciated, for all five tested sensors. Although we provide average resistance ratios for period 1 and period 2 in Table 6, it is difficult to gauge the overall resistance decrease from tabulated values alone. We therefore believe that both figures should be included in the manuscript for sake of comparison.

Finally, we have decided to remove Figure 19, which shows the ratio between measured resistance and modelled baseline resistance for LSCE007, as a function of wind speed. This figure shows that there is no change in resistance ratio with increasing wind speed, which we already suitably summarise in the main test. It is not a key result and is simply a point of discussion. This figure is therefore not a necessity in the main manuscript.

In summary, we have reduced the number of figures in the main manuscript from 19 to 14. We hope that this improves the relevance of the existing figures, whist reducing the overall size of the manuscript. We would like to reiterate our thanks to the Editor for this suggestion and opportunity.

Yours faithfully,

Adil Shah

**Modified manuscript**

[revised manuscript text omitted]